# Drivers of CO₂ emissions during the dry phase in Mediterranean and Temperate ponds

Victoria Frutos-Aragón [1], Sandra Brucet [1,6,9], Rafael Marcé [2], Tuba Bucak [3], Thomas A. Davidson [3], Louisa-Marie von Plüskow [4,5], Pieter Lemmens [5,7], Carolina Trochine [1,8]

[1] Aquatic Ecology Group, University of Vic - Central University of Catalonia, Group, Vic, 08500, Spain
[2] Integrative Freshwater Ecology group, Blanes Centre for Advanced Studies (CEAB-CSIC)
[3] Department of Ecoscience, Aarhus University, Aarhus, Denmark
[4] Department of Fish Biology, Leibniz-Institute of Freshwater Ecology and Inland Fisheries, Berlin, Germany
[5] Laboratory of Freshwater Ecology, Evolution and Conservation, KU Leuven, Leuven, Belgium
[6] Catalan Institution for Research and Advanced Studies (ICREA), 08010 Barcelona, Spain
[7] Research Institute for Nature and Forest (INBO), Brussels, Belgium
[8] Department of Ecology, INIBIOMA CONICET, Universidad Nacional del Comahue, San Carlos de Bariloche, Argentina
[9] Fundació Privada Bionexus, 17003, Girona, Spain

*Correspondence to*: Victoria Frutos Aragón (victoriavfa@gmail.com)

**Abstract.** Pond ecosystems play an important role in the global carbon cycle with the potential to act as both sinks and sources of $CO_2$. However, $CO_2$ emissions during the dry phases of ponds remain underrepresented in global reports, despite growing evidence that climate change-driven shifts in temperature and precipitation are likely to increase the frequency and duration of these dry periods. Here we assess $CO_2$ fluxes from dry pond sediments in relation to climatic region, seasonal changes, and hydroperiod duration. Specifically, we aimed to identify the key environmental drivers shaping $CO_2$ fluxes during the dry phase. We measured $CO_2$ fluxes from bare air-exposed sediments using closed static chambers equipped with internal mini-loggers in 30 ponds across Mediterranean and Temperate regions. Ponds acted as sources of $CO_2$ during dry phases, with emissions ranging from 127 to 4889 mg C $m^{-2}$ $d^{-1}$ (mean ± SD = 1398 ± 1201). Although mean emissions did not differ significantly between climate regions, hydroperiod length interacted with climate and season, showing a significant effect in summer, particularly in Mediterranean ponds, where longer hydroperiods led to higher emissions. Emissions were considerably higher in summer than in autumn, primarily driven by the interaction between sediment temperature and water content. The highest fluxes occurred at approximately 27 ºC and sediment water content between 27% and 44%. Additionally, ponds in better conservation status and with lower carbonate content emitted more $CO_2$. Our findings improve understanding of $CO_2$ emissions during increasingly common dry phases and highlight how climate modulates local sediment conditions, thereby influencing the magnitude of these emissions. This underscores the need for comprehensive assessments of carbon fluxes that incorporate dry-phase emissions, accounting for climate, hydroperiod, and both direct and indirect effects of local environmental drivers.

# 1 Introduction

Inland waters play a fundamental role in the global carbon cycle by processing substantial amounts of organic carbon from terrestrial ecosystems, which can be buried, exported, or released to the atmosphere as $CO_2$ and $CH_4$ (Cole et al., 2007; Raymond et al., 2013; Tranvik et al., 2009). Among these systems, ponds are globally abundant ecosystems (Verpoorter et al., 2014). Despite covering a small fraction of Earth's surface, they exhibit intense biogeochemical processes compared to other freshwater systems (Holgerson and Raymond, 2016). Ponds' $CO_2$ emissions per unit area can be comparable to those of larger freshwater bodies and upland soils (Downing, 2010; Hill et al., 2021; Obrador et al., 2018; Oertli et al., 2009). However, reported $CO_2$ emissions from ponds are highly variable, ranging from hundreds to several thousand mg C $m^{-2}$ $d^{-1}$. For instance, ponds of similar size (< 0.001 and 0.001–0.01 $km^2$) reported by Holgerson and Raymond (2016) emitted on average 254 and 422 mg C $m^{-2}$ $d^{-1}$, respectively, whereas exposed pond sediments reported by Keller et al. (2020) range from -73 to 11765 mg C $m^{-2}$ $d^{-1}$.

This significant variability in reported $CO_2$ fluxes underscores the importance of key factors influencing greenhouse gas (GHG) emissions, such as drying events, which can range from partial water level reductions to complete desiccation (Fromin et al., 2010). Drying events are a common feature experienced by many freshwater systems (Gao et al., 2025; Prananto et al., 2020). Globally, up to 40% of inland waters are considered non-permanent (Pickens et al., 2020), and they are particularly prevalent in arid or semi-arid regions such as the Mediterranean (Sánchez-Carrillo, 2009). The shift between wet and dry phases creates fluctuating anaerobic and aerobic conditions, which affect carbon cycling, particularly $CO_2$ and $CH_4$ dynamics (Downing, 2010; Rulík et al., 2023; Zhao et al., 2020; Zou et al., 2022). As a result, $CO_2$ can dominate the total carbon emissions during dry periods while limiting $CH_4$ (Beringer et al., 2013; Zou et al., 2022).

During wet phases, ponds can act as either sources or sinks of atmospheric $CO_2$, having the potential to store substantial amounts of organic carbon (DelVecchia et al., 2021; Taylor et al., 2019). $CO_2$ fluxes are primarily regulated by diffusion, while their accumulation is controlled by the carbonate buffering system (Cole and Caraco, 1998). In contrast, during dry periods, ponds may be sources of $CO_2$ because the absence of a water column facilitates $CO_2$ diffusion from exposed sediments, frequently resulting in increased emissions (Beringer et al., 2013; Catalán et al., 2014; Gilbert et al., 2017; Obrador et al., 2018). Consistent with these patterns, global comparisons by Zou et al. (2022) and Keller et al. (2020) revealed that $CO_2$ emissions were lower during wet phases than dry phases across temporary inland water ecosystems (streams, reservoirs, wetlands, lakes, and ponds) spanning various climate regions. Particularly, ponds exhibited the highest sediment-driven emissions across inland freshwater ecosystems (Keller et al., 2020).

Previous studies suggest that $CO_2$ emissions from ponds during the dry phases are primarily governed by the same drivers that regulate emissions in natural soils and other aquatic environments (Håkanson, 1984; Keller et al., 2020; Martinsen et al., 2019; Oertel et al., 2016). The primary source of $CO_2$ efflux is the biological activity on organic matter in the sediments. Increased oxygen availability during dry conditions stimulates enzymatic activity and microbial degradation of organic matter by bacteria and fungi, enhancing $CO_2$ release (Fromin et al., 2010). Local factors strongly influence this process,

primarily depending on sediment temperature, sediment water content, and organic matter concentration (Agnew et al., 2021; Fraser et al., 2016; Jarvis et al., 2007; Lloyd and Taylor, 1994; Pozzo-Pirotta et al., 2022; Suh et al., 2009). Additionally, the composition of organic matter (e.g., humic-like components) further affects the $CO_2$ emissions (Catalán et al., 2014; Obrador et al., 2018). Other edaphic factors, such as porosity, structure, bulk density, pH, and the chemical and biological characteristics, can also modulate the $CO_2$ emissions (Buragienė et al., 2019). Among biological components, macrophytes influence $CO_2$ emissions by altering sediment conditions and microbial activity (Baastrup-Spohr et al., 2016; Tak et al., 2023; Weise et al., 2016). They contribute nutrients and organic compounds, including fulvic acids, which are readily decomposed by microorganisms, thereby enhancing the respiration and decomposition of organic matter (Bottino et al., 2021; Wang et al., 2017).

Regional meteorological conditions, such as air temperature and precipitation, strongly influence local and edaphic factors linked to $CO_2$ emissions. These emissions respond sensitively to changes in sediment temperature (cold–heat) and water content (drought–excess water) (Fromin et al., 2010; Unger et al., 2010). Rewetting of previously dry soils, driven by precipitation, can trigger a rapid increase in microbial activity and $CO_2$ release. This phenomenon is known as the "Birch effect" (Fromin et al., 2010; Jarvis et al., 2007). In addition, land use and conservation status can indirectly affect emissions by modifying nutrient loading in sediments and the inflow of organic matter (Bartrons et al., 2025; Novikmec et al., 2016). These factors are known to affect emissions during the wet phase (Bhushan et al., 2024; Morant et al., 2020). However, their potential role in shaping emissions during the dry phase remains largely underexplored.

The combined effects of climate change, land-use and water abstraction are likely to intensify the frequency and severity of drying events in regions worldwide, including central Europe and the Mediterranean region (Burkett and Kusler, 2000; Dimitriou et al., 2009; IPCC, 2014; Oroud, 2024; Pekel et al., 2016; Voudouri et al., 2023; Williams et al., 2010). In this context, it is especially concerning that the dry phases of temporary inland water bodies are often excluded in GHG studies, which tend to focus predominantly on wet periods, overlooking the significant role that dry sediments can play in carbon cycling (Marcé et al., 2019). This knowledge gap likely underestimates the impact of hydrological dynamics, severely limiting our understanding of the key drivers of $CO_2$ emissions during the dry phases (Hanson et al., 2015; Keller et al., 2020; Premke et al., 2016). Furthermore, the sequence and duration of wet and dry periods can create ecological and biogeochemical feedbacks, where the length of each phase influences the next, thereby modulating carbon processing, macrophytes development, and greenhouse gas emissions throughout the hydroperiod (Girard et al., 2024; Jin et al., 2023). For this reason, our study aims to address this gap by identifying the main drivers of $CO_2$ fluxes during dry periods and examining how the preceding wet phase, in terms of hydroperiod length (i.e., the duration of water presence prior the dry phase in a pond throughout the year) influence them, through a comparison of ponds from contrasting climatic regions across two seasons.

Here, we quantified $CO_2$ fluxes from air-exposed pond sediments during the dry phase across different seasons (summer and/or autumn) in 30 ponds with varying hydroperiod lengths, located in Mediterranean and Temperate climate regions.

Within this environmental and biogeochemical framework, we aimed to identify the key drivers regulating $CO_2$ emissions from dry pond sediments.

Specifically, we asked: (1) How do seasonal and climatic differences influence $CO_2$ fluxes from pond sediments during the dry phase?; (2) To what extent do climate and hydroperiod length affect $CO_2$ emissions under dry conditions?; (3) Which local sediment and environmental variables, including possible indirect factors, best explain variation in $CO_2$ fluxes across ponds?; (4) Does macrophyte coverage, potentially reflecting conservation status, influence $CO_2$ emissions during the dry phase?. Based on these questions, we hypothesized that: (1) $CO_2$ fluxes from sediments during the dry phase will vary across seasons and climate regions, with higher emissions in summer due to enhanced microbial activity and organic matter decomposition; (2) climate and hydroperiod length will influence $CO_2$ emissions, with shorter hydroperiods (ponds that are inundated for shorter periods) leading to lower emissions due to reduced sediment water content and organic matter availability; (3) local sediment conditions such as temperature, sediment water content, and organic matter will primarily drive emissions, while climate sets broader seasonal and hydrological patterns; and (4) ponds with larger macrophyte coverage, reflecting better conservation status (e.g., clear water with turbidity < 5 NTU, extensive native emergent vegetation, and ≥ 50% hydrophytic plant cover, particularly vascular submerged species or charophytes covering > 75% of the pond bottom), will exhibit greater $CO_2$ emissions due to increased vegetation senescence during the dry phase.

## 2 Methodology

### 2.1 Pond selection and data collection

We collected data from 30 ponds across four countries in Europe distributed in two climate regions, Temperate ($N = 14$) (Belgium, Denmark and Germany) (Fig. 1a) and Mediterranean ($N = 16$) (Spain) (Fig. 1b), spanning latitudes from 41° 49' 9" to 54° 54' 8" N and longitudes from 2° 21' 7" to 14° 21' 9" E (Table A1).

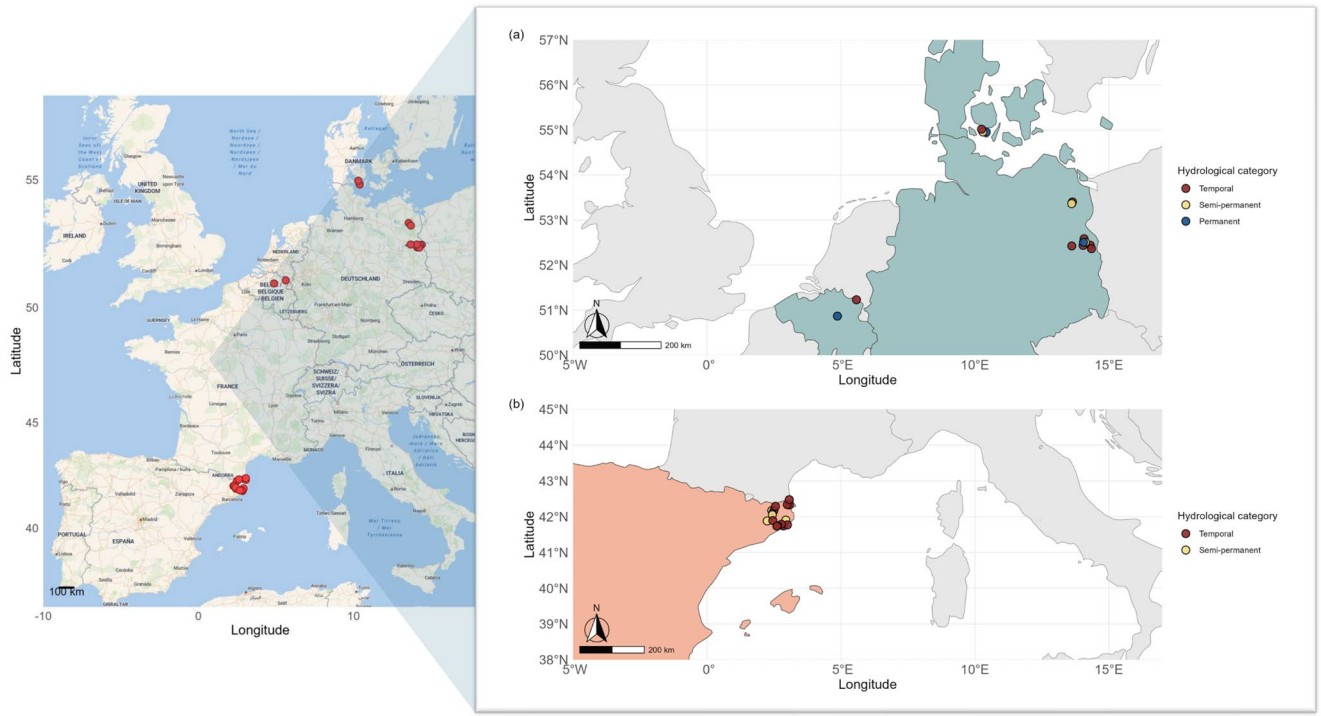

**Figure 1: Geographic locations of the studied ponds. Left panel: map created using Stadia Maps outdoors basemap and**
120 **OpenStreetMap data. Map data © OpenStreetMap contributors, © Stadia Maps (https://stadiamaps.com/). Ponds are highlighted in red. Right panel: map showing ponds categorized by hydroperiod: temporary (dark red), semi-temporary (yellow), and permanent (dark blue). Countries are colour-coded according to their climate regions: Mediterranean (orange) and Temperate (light blue).**

Our study included 20 temporary, 5 semi-permanent, and 3 permanent ponds. Temporary ponds dried completely every year,
semi-permanent ponds dried some years (including the sampling year), and permanent ponds never fully dried. We based this classification on hydrological information obtained from field monitoring and logger data collected between 2021 and 2023. During 2022 (the sampling year), these ponds either dried out or experienced significant water level declines (e.g., permanent ponds), exposing large areas of the pond basin sediments to air (on average, $84 \pm 20\%$ of the total pond area). All ponds were in lowland rural areas below 800 m elevation, spanning diverse land uses such as cattle-grazing lands,
agricultural fields, and protected nature reserves (López-de Sancha et al., 2025). The selected ponds covered a wide range of hydro-geomorphological and sediment characteristics (Table A2). We measured $CO_2$ fluxes from exposed bare sediment during summer ($N = 23$) and/or autumn ($N = 23$), including ponds in one or both seasons, depending on the timing and extent of their water level decline. We used a combination of regional and local factors to characterize the ponds and their surrounding landscapes, aiming to identify the main drivers of dry sediment $CO_2$ fluxes (Table A3):

## 2.2 Environmental and landscape variables

### 2.2.1 Climatic regional data

We used temperature (40-year mean of annual temperatures (1978-2018) and the annual mean for 2022) and precipitation (40-year mean annual precipitation (1978-2018) and the annual mean for 2022) calculated from the nearest cell 1 X 1 km to each pond from the fifth generation of ECMWF atmospheric reanalysis of the global climate (ERA5) Copernicus Climate Change Service (C3S) (Wouters, 2021). We used long-term climatic averages (1978-2018) to represent the average climatic condition in the region, and the mean annual values for 2022 to capture the exceptional heat and drought conditions during the sampling year.

### 2.2.2 Hydrological data

We defined hydroperiod length as the number of months with surface water present during the 12 months preceding the final sampling (autumn), representing the duration of the preceding wet phase. We used hydroperiod length because it is a simple, easily obtained metric that indirectly integrates a suite of biotic and abiotic processes occurring during the preceding wet phase, such as carbon processing and macrophyte development. We estimated hydroperiod length using temperature data (HOBO U2OL-0X, Onset) and/or water level loggers (Cera-Diver, Schlumberger). The water level loggers operate based on differential pressure measurements, enabling the detection of water presence. For temperature-based assessments, we deployed two loggers: one outside the pond and another at its maximum depth. The pond was considered dry when the recorded temperatures from both sensors coincided, indicating exposure of the deepest zone. We measured the pond depth profile *in situ* along two perpendicular transects at 2 m intervals using a graduated pole perpendicular to the base ground of the pond, to calculate the maximum depth. We included maximum depth in the models to assess whether this geomorphological trait (shallowness) could indirectly influence $CO_2$ emissions under dry conditions. Pond areas were delineated through manual polygon digitization in Google Earth Pro (Google LLC, 2021), enabling georeferenced surface area calculations suitable for spatial ecological analysis.

### 2.2.3 Water physicochemical data

We measured physicochemical parameters and chlorophyll a during the preceding wet phase (spring) and used them as proxies for the ponds' trophic status. We determined total nitrogen (TN) and total phosphorus (TP) concentrations ($mgL^{-1}$) from unfiltered water samples (250 ml) following Sen Gupta and Koroleff (1973). We measured dissolved organic matter (DOC) in mg $L^{-1}$ after filtering water samples through pre-combusted GF/F filters (0.7 μm pore size), using a TOC-L analyzer (TOC-L, Shimadzu). We measured chlorophyll a (μg $L^{-1}$) using an Algae fluorometer (AlgaeTorch, bbe Moldaenke). For more detailed information, see Bartrons et al. (2025).

### 2.2.4 Landscape data

We classified six land-use types: open nature, forest, pasture, arable, grassland and urban. Land use was assessed within 5 m and 100 m radii from each pond. We estimated the percentage cover of each land-use type based on visual field observations beyond these radii.

### 2.2.5 Conservation data

We calculated the ECELs index (Ecological Conservation of Ephemeral Lentic Systems (Sala et al., 2005)) as a measure of
pond conservation status. This index is based on five main components of the pond: pond morphology (e.g., littoral slope, burial, impermeabilization), human impacts around the pond (e.g., infrastructure, nearby land uses, protected area), water aspects (e.g., transparency, odour), emergent vegetation (e.g., perimeter and in-pond cover, dominant community), and hydrophytic vegetation (e.g., submerged or floating vegetation). This index ranges from 0 to 100, with higher values indicating better pond conditions. In this study, we analyzed a subset of ECELS values previously reported in López-de
Sancha et al. (2025a; 2025b) to assess their potential relationship with $CO_2$ emissions during the dry phase.

### 2.2.6 Macrophyte data

We measured macrophytes coverage *in situ* as the percentage of the pond's area covered by emerged vegetation. We also calculated macrophytes PVI (Plant Volume inhabited) as the percentage of water volume occupied by macrophytes (submerged, floating, and emergent) in the pond.

**2.3 $CO_2$ dry flux measurements**

We measured $CO_2$ fluxes from the surface sediment using closed static chambers in a total of 30 ponds during summer (July-August) and/or autumn (September-November) of 2022. Measurements were conducted once per season at each pond during daytime (08:00–19:00 h), with no temporal replicates within the same season (Table A1; S1).

$CO_2$ fluxes were measured by monitoring concentration changes over time using mini-loggers placed in static chambers. The
chambers consisted of opaque polypropylene buckets covered with aluminium tape to minimize solar warming of the headspace. Each chamber had an internal volume of 8 L (diameter 345 mm; height 160 mm) and enclosed a sampling area of 0.075 m². The mini-loggers inserted in the chambers were built following the methodology described by Bastviken et al. (2015). We used a $CO_2$ sensor module (SCD30, Sensirion) to log $CO_2$, temperature, and humidity every 2-4 seconds. The sensor was connected to an electronic circuit board (Mega250, Arduino or Uno Arduino) equipped with a real-time clock, an
SD card for data storage, and powered by an external battery (RS PRO Q2). The chamber had a small fan inside at the top to recirculate the air and prevent stratification, but there was no airflow through the chamber.

To account for the heterogeneity of the pond basin, we selected between four and eight spots in each pond, depending on its size, to measure the $CO_2$. The chambers were placed on bare sediment and fitted to plastic collars inserted approximately 1

cm into the soil to minimise lateral gas leakage. We measured $CO_2$ fluxes for 5 minutes at all spots in each pond, with a 5-minute flushing interval. At one spot per pond, we conducted a 1-hour closing time measurement using the same internal $CO_2$ sensor. Logger data and integration were comparable to the 5-minute measurements. During the 1-hour measurement, we collected six 10 mL manual gas samples every 10 minutes (using a 60 mL syringe, BD Plastipak, and 5.9 mL vacuum Exetainer® vials, Labco) to assess the reliability of sensor-based measurement.

The $CO_2$ sensors were pre-calibrated by the manufacturer (Sensirion AG, 2020) and provided data in ppm units. The sensors were configured following manufacturer recommendations, including forced recalibration (410 ppm), altitude compensation (60 m), atmospheric pressure compensation (1000 mbar), and a fixed measurement interval, ensuring stable operation across deployments. Measured concentrations were corrected for water vapour using the simultaneously recorded temperature and relative humidity data. To calculate $CO_2$ fluxes, we applied a 3-point rolling average to the raw data to reduce background noise. The initial 1-2 minutes of each measurement period were excluded due to increased signal noise likely caused by humidity and temperature fluctuations immediately after chamber closure. Flux values were then derived from the data corresponding to the last 2-3 minutes of each 5-minute measurement period to ensure more accurate linear flux estimates (Johannesson et al., 2024). For the 1-hour measurements, the same approach was applied but using longer time intervals within each recording to obtain the linear fit. The $CO_2$ flux was calculated by the ideal gas law using the following Eq. (1) (Podgrajsek et al., 2014):

$$F_{dif} = \left( \frac{\Delta C_i}{\Delta t} \cdot \frac{P \cdot M}{R \cdot T} \cdot \frac{Vi}{Ai} x\ 1000 \right) x\ (60\ x\ 60\ x\ 24) \qquad (1)$$

Where $F_{dif}$ represents the diffusive flux (mg C m$^{-2}$ d$^{-1}$), $\frac{\Delta C_i}{\Delta t}$ is the change in gas concentrations (ppm·10$^{-6}$), P is the atmospheric pressure (atm), R is the gas constant (L·atm/mol·K), T is the temperature (K), M is the molar mass of carbon (g mol$^{-1}$), $Vi$ is the volume of the chamber (L), $Ai$ is the area of the chamber (m$^2$). The factor of 1000 accounts for the unit conversion from grams to milligrams, and the multiplication 60 x 60 x 24 accounts for the conversion from seconds to days. The measurement approach used in this study follows Bastviken et al. (2015), who provide detailed information on logger preparation, sensor evaluation, calibration, and data processing in their manuscript and supplement.

## 2.4 Sediment characterisation

After measuring $CO_2$, we collected sediment samples from the upper 5 cm of the surface layer and recorded temperature (°C) *in situ* using a portable soil probe thermometer (30.1033, TFA Dostmann). In the laboratory, we measured pH (HI 98127, HANNA) and conductivity (BiXs50, Violab), in µS cm$^{-1}$ at 25 °C in a 1:2.5 sediment: ultrapure water (Milli-Q) mixture. Sediment water content (%) was determined as weight loss after drying 5 g of fresh homogenized sediment at 105 °C furnace muffle (Carbolite Gero, CWF 12/13) for 48h. Organic matter (%) and carbonate content (%) in the sediments were determined through sequential loss on ignition in the muffle furnace and expressed as percentages of the dry sediment weight. Samples were combusted at 500 °C for 4 hours to estimate organic matter content, followed by combustion at

950 °C for 2 hours to assess mass loss associated with carbonates, commonly used as a proxy for carbonate mineral content (Heiri et al., 2001; Martinsen et al., 2019). We assessed sediment texture using a hand-texturing method, categorized samples on the dominant material, and grouped them as clay, loamy or sandy (Thien, 1979).

### 2.4.1 Dissolved organic matter characterization

To assess the composition of organic matter in sediments, we extracted the water-extractable organic matter (WEOM) in a 1:40 sediment water ratio (p/p) following Obrador et al. (2018). We used the solutions obtained from WEOM to determine the dissolved organic carbon (DOC) and characterized dissolved organic matter (DOM) in the sediment. We analyzed DOC using a TOC analyzer (TOC-VCS, Shimadzu) and expressed the results as mg C g$^{-1}$. We characterized DOM using absorbance and fluorescence spectroscopy to gain insights into the composition, origin, and reactivity of sedimentary organic matter. We obtained the UV-Vis absorbance spectra (200–800 nm) using a UV-Vis spectrophotometer (Cary 4000, Agilent) and fluorescence excitation-emission spectra using a fluorescence spectrophotometer (F-700, HITACHI). From DOM characterization, we calculated several absorbance and fluorescence indices: 1) the humification index (HIX; unitless), which indicates the degree of organic matter humification, as the ratio of the fluorescence emission peak areas between 435–480 nm and 300–345 nm at an excitation wavelength of 254 nm. 2) The biological index (BIX; unitless), which reflects the autochthonous productivity and freshness, as the ratio of the fluorescence intensities between 380 and 430 nm at an excitation wavelength of 310 nm (Fellman et al., 2010; Gabor et al., 2014; Huguet et al., 2009). 3) The fluorescence index (FI; unitless), used to assess the proportion of autochthonous versus allochthonous organic matter, as the ratio of emission intensities between 4750-500 nm at an excitation wavelength of 370 nm. 4) Specific ultraviolet absorbance (SUVA; L mg$^{-1}$ m$^{-1}$), an indicator of aromaticity, calculated by dividing the UV coefficient absorbance at 254 nm by the DOC concentration (mg/L) (Weishaar et al., 2003). 5) Absorbance at 254 nm, indicating the presence of aromatic organic compounds and used to assess the quantity and complexity of DOM. 6) Absorbance at 300 nm, reflecting the amount of less complex and more biodegradable organic compounds.

We further analyzed the DOM fluorescence properties using parallel factor analysis (PARAFAC) following Murphy et al. (2013). Multiple models were evaluated to determine the optimal number of components, based on split-half analysis validation, core consistency, model fit, and residual examination. We used the Openfluor.org platform to identify fluorescent components by matching excitation and emission spectra to models in the repository, with a Tucker Congruence Coefficients (TCC) accuracy of 0.99.

We processed DOM and PARAFAC data using the R package StaRdom (Pucher et al., 2019). Further methodological details and processing steps are available in the supplementary material.

**2.5 Statistical analysis**

We used the non-parametric Mann–Whitney test (Wilcoxon rank-sum test) to compare overall $CO_2$ fluxes across climates (Mediterranean, Temperate) and seasons (Summer, Autumn) (Hypothesis 1), based on the average of multiple measurements in each pond (wilcox_test function in rstatix package (Kassambara, 2023)). Before analysis, data were assessed for normality using Shapiro–Wilk tests (shapiro.test function in stats package (R Core Team, 2024)). To assess whether our results significantly differed from published values of pond dry fluxes, we conducted Welch's t-tests (t.test function in stats

package) based on summary statistics (means, standard deviations, and sample sizes).

To investigate the influence of hydroperiod length on $CO_2$ fluxes, we applied a linear mixed-effects model (LMM) (lmer function in lme4 package (Bates et al., 2015)). This model tested whether the slopes and intercepts describing the relationship between hydroperiod length and $CO_2$ emissions differed significantly between climates and seasons. Pond ID was included as random effect (Bolker et al., 2009), with $CO_2$ emissions as the response variable and hydroperiod length,

season, and climate as the fixed effects. We calculated marginal and conditional R-squared values for the models (r.squaredGLMM function in MuMIn package (Bartoń, 2023)). To evaluate the significance and direction of the effect of hydroperiod length within each climate and season, we extracted the estimated slopes and their 95% confidence intervals (CI) from the model (emtrends function in emmeans package (Lenth, 2025)). This approach allowed us to assess how the effect (slope) of hydroperiod length on $CO_2$ emissions varied across seasons and climate regions (Hypothesis 2).

Then, to identify the main drivers of fluxes across climate regions, seasons and hydroperiod (Hypotheses 3 and 4), we used Generalized Linear Mixed Models (GLMMs) using a Gaussian distribution. For these analyzes, we treated all subsamples as individual observations in the GLMMs, including Pond ID as a random intercept to account for nested data structure. We normalized all numeric variables (predictors and response) using Ordered Quantile (ORQ) normalization (orderNorm function in bestNormalize package (Ryan A. Peterson, 2021)), a one-to-one transformation that transforms values into a

vector that follows a Gaussian distribution (Peterson and Cavanaugh, 2020).

We assessed correlations among predictor variables (Table A3) using Pearson's correlation (Cor function in corrplot package (Wei and Simko, 2024)), retaining variables with low multicollinearity ($|r| < 0.6$) as fixed effects in the GLMMs. We fitted GLMMs (glmer function in lm4 package (Bates et al., 2015)), with $CO_2$ fluxes as the response variable and Pond ID as a random intercept (Bolker et al., 2009). We also tested biologically relevant interactions, including the one between sediment

temperature and sediment water content (Fig. S1). To capture the nonlinear relationship between sediment water content and $CO_2$ fluxes, we included second-degree polynomial terms (using orthogonal polynomials) (poly function in stats package ) in our GLMMs (Fox, 2003). This approach allowed us to model both the linear and quadratic effects of sediment water content on $CO_2$ fluxes while minimizing multicollinearity. We used multi-model inference, an automated model selection (dredge function in MuMIn package) to identify the best models based on Akaike's Information Criterion (AIC) (Akaike, 1998). We

calculated marginal and conditional $R^2$ values for the models (r.squaredGLMM function in MuMIn package). While p-values, coefficients, and standard errors of fixed effects were obtained directly from the model (sjPlot package (Lüdecke,

2021)). We applied a four-step filtering approach to select the final models. First, we retained models in which all predictor variables were statistically significant ($p < .05$). Second, we assessed multicollinearity using the Variance Inflation Factor (VIF) (vif function in car package (Weisberg and Fox, 2011)), ensuring that all variables included had VIF < 5. Third, we evaluated the model fit and validity using residual diagnostics (TestDispersion and testUniformity functions in DHARMa package (Hartig, 2022)). This last step included assessing homoscedasticity, normality, and the absence of overdispersion or spatial autocorrelation in the residuals. Finally, we selected the model with the lowest AIC value (Akaike, 1998).

All statistical analyses were performed using R Studio software version 4.4.2 (R Core Team, 2024). We used the ggplot 2 R package (Wickham, 2016) to create the plots and ggmap R package for maps (Kahle and Wickham, 2013) and Visreg for partial plots of GLMMs (Breheny and Burchett, 2017).

## 3 Results

Our study showed that ponds were on average sources of $CO_2$ to the atmosphere during the dry period, with $CO_2$ sediment emissions ranging from 127 to 4889 mg C m$^{-2}$ d$^{-1}$ (mean ± SD: 1398 ± 1201, median = 1078, $N = 30$). However, we also recorded some negative values indicating localized $CO_2$ uptake from the atmosphere, accounting for about 4% of the total fluxes measured. $CO_2$ fluxes exhibited high variability both among ponds and within ponds (Fig. 2; Table S1).

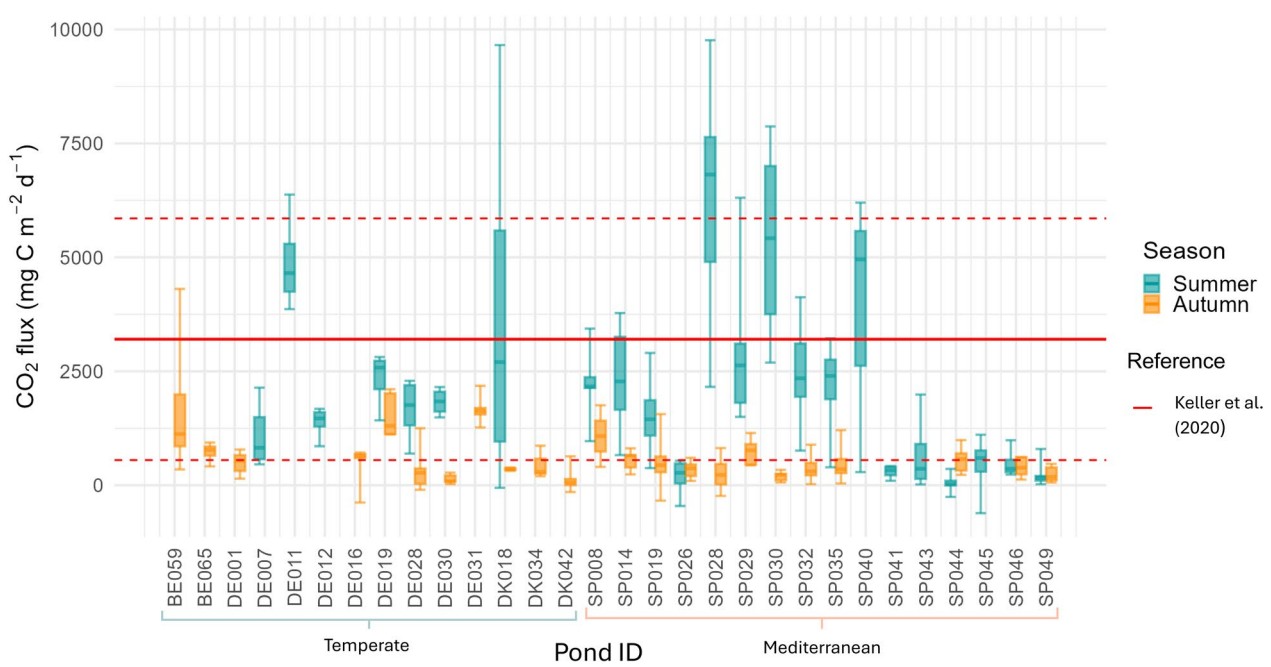


**Figure 2. Boxplots of CO$_2$ fluxes for each pond in summer and autumn. Boxes represent the interquartile range with the median indicated by a solid line; outliers are shown as individual points. The red lines indicate the CO$_2$ fluxes reported by the study on dry fluxes on ponds by Keller et al. (2020), the red solid line indicates the mean, and the red dashed lines represent the standard deviations. Pond IDs are grouped using brackets according to their Climate region (Temperate or Mediterranean). The first two capital letters indicate the country initials (see Table A1).**

The studied ponds exhibited significant climatic and seasonal variability in their hydro-geomorphological and sediment characteristics (Table A2, S2). Mediterranean ponds exhibited higher air and sediment temperatures. They also showed lower sediment water content and reduced macrophyte coverage (Table S2). In contrast, Temperate ponds exhibited lower air and sediment temperatures, higher sediment water content, and greater macrophyte coverage (Table S2). Seasonal variation was significant only for sediment temperature (U = 0, $n_1$ = 23, $n_2$ = 23, $p$ < .001, r = .86). Regarding sediment characteristics, PARAFAC analysis of fluorescent DOC yielded a three-component model. The components were expressed as the percentage contribution of each component to the total fluorescent DOC. We identified the main components as: component 1 (C1), corresponding to terrestrial humic-like substances, component 2 (C2), corresponding to humic-like substances, and component 3 (C3), corresponding to tryptophan-like substances (Table S3–S5). Considering sediment composition, Mediterranean ponds exhibited lower organic matter and DOC but higher content of C2 (humic-like) opposite to Temperate ponds (Table S2). Consistent with wet phase physicochemical patterns, Mediterranean ponds showed lower concentrations of TN, TP, and chlorophyll a than Temperate ponds. Temperate ponds also had a higher percentage of pasture within 5 m and 100 m buffer zones and a larger proportion of urban area within the 100 m buffer (Table S2).

## 3.1 Climate, season, and hydroperiod variation in CO$_2$ emissions

CO$_2$ emissions from the Mediterranean ponds ranged from 242 to 3877 mg C m$^{-2}$ d$^{-1}$ (mean ± SD: 1394 ± 1207, median = 1212, $N$ = 16), while emissions from Temperate ponds ranged from 127 to 4889 mg C m$^{-2}$ d$^{-1}$ (mean ± SD: 1403 ± 1239, median = 1078, $N$ = 14). Fluxes did not differ between climate regions (U = 104, $n_1$ = 16, $n_2$ = 14, $p$ = .76, r = .06; Fig. 3). However, we found significant seasonal differences in fluxes (U = 119, $n_1$ = 23, $n_2$ = 23, $p$ = .001, r = .47; Fig. 3). During summer, fluxes ranged from 46 to 6289 mg C m$^{-2}$ d$^{-1}$ (mean ± SD: 2111 ± 1739, median: 1831, $N$ = 23), whereas in autumn, fluxes were significantly lower, ranging from 127 to 1726 mg C m$^{-2}$ d$^{-1}$ (mean ± SD: 598 ± 464, median: 452, $N$ = 23).

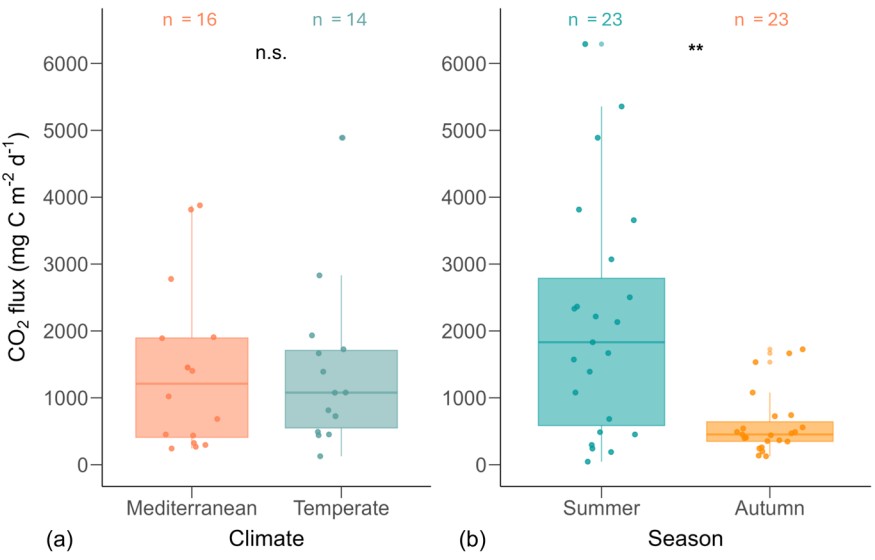

**Figure 3. Boxplots of CO₂ fluxes by climate (a) and season (b). Each point represents the mean CO₂ flux per pond, calculated from multiple replicate measurements. Asterisks (\*\*) indicate statistically significant differences based on Mann–Whitney U tests (p < .01); n.s. denotes non-significant differences. The number of ponds (n) is indicated above each boxplot. Boxplot structure is described in Figure 2.**

The linear mixed-effects model (LMM) assessing the influence of hydroperiod length across season and climate regions on $CO_2$ fluxes revealed a significant three-way interaction among season, climate and hydroperiod ($R^2_m$ = .33, $R^2_c$ = .58, $p <$ .01). This indicates that the effect of hydroperiod was season-specific and climate-dependent (Fig. 4). Specifically, hydroperiod length significantly influenced $CO_2$ emissions only during summer in both climate regions ($p < .001$). In Mediterranean ponds, longer hydroperiods were associated with increased $CO_2$ emissions ($p < .001$), whereas in Temperate ponds, the trend was inverse but not statistically significant ($p < .13$) (Table 1).

**Figure 4. Relationship between CO₂ fluxes and hydroperiod length (months) by climate and season. Points represent individual ponds. Solid lines show linear regressions by season (blue = summer; orange = autumn) with 95% confidence intervals shaded. Marginal and conditional R² values and p-value are shown within the plot.**

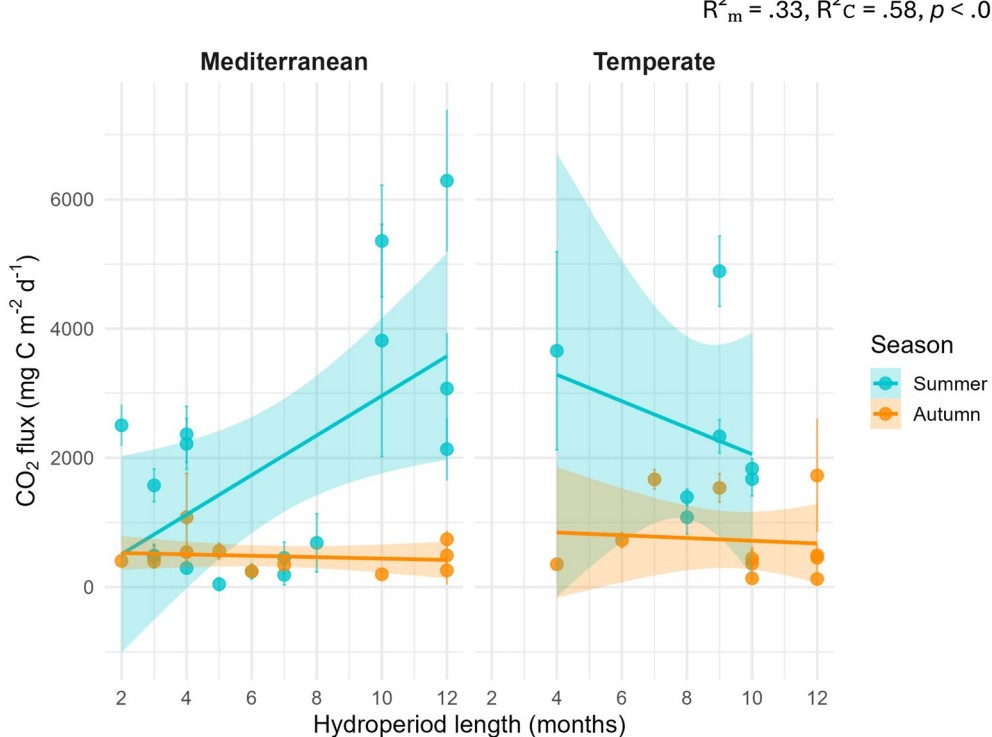

**Table 1. Estimated slopes (effects) of hydroperiod length on CO$_2$ fluxes for each combination of season (summer and autumn) and climate region (Mediterranean and Temperate). Values are derived from the linear model assessing the interaction between hydroperiod, season, and climate. Abbreviations: SE = standard error; df = degrees of freedom; CL = confidence limit; CI = confidence interval.**

| Climate | Season | Estimate | SE | df | lower.CL | upper.CL | p-value |
|---------|--------|----------|------|------|----------|----------|---------|
| Mediterranean | Summer | 290.82 | 78.69 | 27.80 | 129.58 | 452.06 | < .001 |
| Mediterranean | Autumn | 17.98 | 81.67 | 31.40 | -148.49 | 184.46 | .83 |
| Temperate | Summer | -264.01 | 172.97 | 68.92 | -609.09 | 81.08 | .13 |
| Temperate | Autumn | -17.04 | 131.82 | 38.03 | -283.89 | 249.81 | .90 |

We also explored the relationship between CO$_2$ fluxes and temperature (annual and 40-year mean), a variable that displayed a moderately strong inverse correlation with hydroperiod length (Fig. B1). Despite this correlation, the overall regression model encompassing emissions from all ponds, and the regressions conducted for each climate region, did not yield a significant relationship ($p > .05$).

## 3.2 Drivers of CO$_2$ fluxes

The variables identified as potential predictors of CO$_2$ fluxes were sediment water content, sediment temperature, hydroperiod length, conservation status, macrophyte coverage, PVI, pH, maximum depth, conductivity, carbonate content, DOC, BIX, HIX, and SUVA, which exhibited correlation coefficients ($r$) predominantly between 0.6 and -0.6 (Fig. B1; B2). The remaining variables were excluded due to high collinearity ($|r| > 0.6$) or lack of significant association with CO$_2$ fluxes during the dry phase, in the case of land-use and water physicochemical variables. Although DOC components (C1, C2, C3) were excluded from the analysis because of their strong correlation with fluorescence indices (FI, HIX, and BIX), they provide complementary insights into DOC transformation and degradation patterns, presented in the supplementary material (Figs. S2–S5).

The GLMM analysis revealed significant effects of sediment water content, sediment temperature, pond conservation status, carbonate content and the interaction between sediment temperature and water content (Table 2). The model only including the interaction between sediment temperature and water content, had a marginal R$^2$ of 0.35, highlighting the importance of these factors in explaining CO$_2$ fluxes. Additionally, incorporating the second-degree polynomial improved our model by lowering the AIC and increasing the marginal R$^2$ to 0.38, thereby better capturing the curvilinear relationship between water content and temperature (Fig. 5). At lower temperatures (e.g., ~9.4 °C), CO$_2$ emissions increased slightly with sediment water content up to approximately 44%, beyond which the fluxes declined. At intermediate temperatures (~18.2 °C), sediment water content had a moderate effect on CO$_2$ fluxes. In contrast, at higher temperatures (~27.7 °C), the trend reversed, and greater sediment water content led to increased CO$_2$ emissions. However, beyond ~50% water content, this effect could not be reliably assessed: confidence intervals widened substantially, reflecting the lack of observed data in this

range and the consequent extrapolation of model predictions (Fig. 5). Finally, the $CO_2$ fluxes showed a positive relationship with conservation status and a negative relationship with carbonate content (Table 3).

375
**Table 2. Model comparison of generalized linear mixed models (GLMMs) explaining $CO_2$ fluxes during the dry phase in ponds. The best model, selected based on AIC (Akaike Information Criterion), is highlighted in bold. Abbreviations: BIC = Bayesian Information Criterion; df = degrees of freedom. Temperature refers to sediment temperature.**

| Model fixed effects | AIC | BIC | Deviance | Log-likelihood | Marginal $R^2$ | Conditional $R^2$ | N | df |
|---|---|---|---|---|---|---|---|---|
| None (null model) | 684.92 | 695.47 | 678.92 | -339.46 | NA | .266 | 249 | 3 |
| Temperature | 655.71 | 669.78 | 647.71 | -323.85 | .123 | .429 | 249 | 4 |
| Water content x Temperature | 604.55 | 625.65 | 592.55 | -296.27 | .347 | .515 | 249 | 6 |
| (Water content + Water content $^2$) x Temperature | 585.99 | 614.14 | 569.99 | -284.99 | .378 | .525 | 249 | 8 |
| (Water content + Water content $^2$) x Temperature + Conservation status | 583.81 | 615.47 | 565.81 | -282.91 | .416 | .520 | 249 | 9 |
| (Water content + Water content $^2$) x Temperature + Carbonate content | 581.27 | 612.89 | 563.27 | -281.63 | .392 | .554 | 248 | 9 |
| (Water content + Water content $^2$) x Temperature + Conductivity | 584.94 | 619.60 | 566.94 | -283.45 | .403 | .545 | 249 | 9 |
| (Water content + Water content $^2$) x Temperature + Conservation status + Conductivity | 584.65 | 619.82 | 564.64 | -282.32 | .431 | .545 | 249 | 10 |
| **(Water content + Water content $^2$) x Temperature + Conservation status + Carbonate content** | **578.43** | **613.56** | **558.43** | **-276.15** | **.431** | **.543** | **248** | **10** |
| (Water content + Water content $^2$) x Temperature + Conservation status + Carbonate content + Conductivity | 578.76 | 617.41 | 556.76 | -278.38 | .447 | .555 | 248 | 11 |

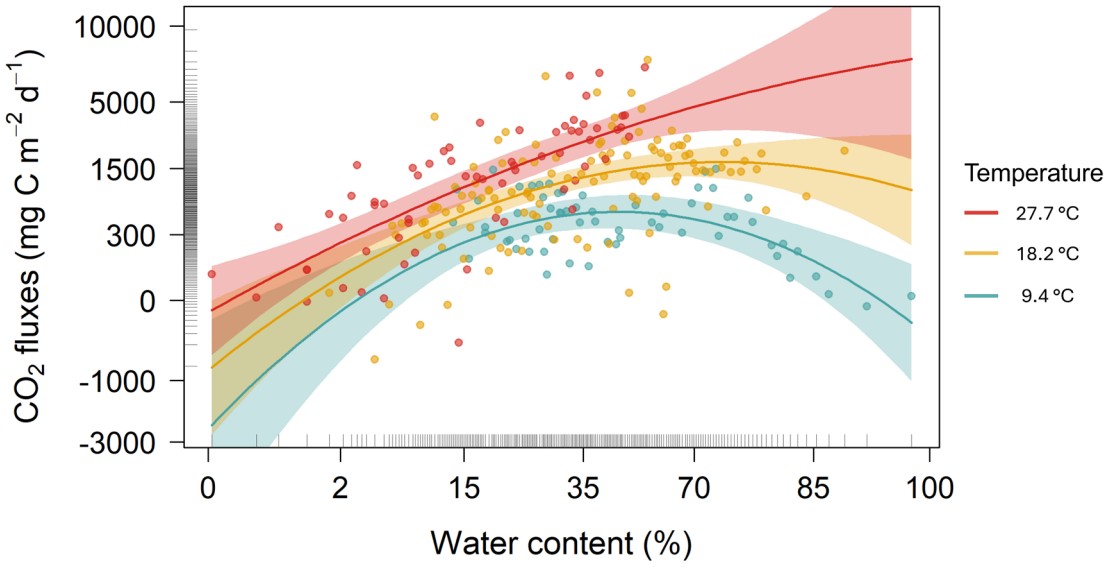

**Figure 5. Partial plot effects of the interaction between sediment water content and sediment temperature on $CO_2$ fluxes. The figure shows results from a Generalized Linear Mixed Model (GLMM) with axes back-transformed from ORQ (Ordered Quantile normalization) to the original scale. Each dot represents model-adjusted values, coloured lines and shaded areas depict fitted trends and 95% confidence intervals at three temperature levels: blue = 9.4 °C, yellow = 18.2 °C, and red = 27.7 °C**

**Table 3. Results of the best Generalized Linear Mixed Model (GLMM) explaining $CO_2$ fluxes during the dry phase of the studied ponds. The Model: (Water content + Water content$^2$) × Temperature + Conservation status + Carbonate content (random intercept for Pond). The table shows predictors, parameter estimates, 95% confidence intervals (CI), and p-values.**

|  | CO₂ fluxes | | |
|---|---|---|---|
| *Predictors* | *Estimates* | *CI* | *p-value* |
| (Intercept) | 0.07 | -0.10 – 0.24 | .407 |
| Water content [Linear] | 8.32 | 5.54 – 11.10 | **< .001** |
| Water content $^2$ [Quadratic] | -4.30 | -6.90 – -1.70 | **.001** |
| Temperature | 0.54 | 0.41 – 0.66 | **< .001** |
| Conservation status | 0.25 | 0.09 – 0.42 | **.003** |
| Carbonate content | -0.20 | -0.33 – -0.08 | **.002** |
| Water content [Linear] x Temperature | 2.69 | 0.33 – 5.04 | **.025** |
| Water content $^2$ [Quadratic] x Temperature | 2.12 | 0.47 – 3.76 | **.012** |

| **Random Effects** | |
|---|---|
| $\sigma^2$ | 0.49 |
| $\tau_{00\ Pond}$ | 0.12 |
| ICC | 0.20 |
| N $_{Pond}$ | 30 |
| Observations | 248 |
| Marginal R$^2$ / Conditional R$^2$ | .431 / .543 |

## 4 Discussion

Our study demonstrated that, overall, ponds emitted $CO_2$ during the dry phase, despite occasional negative fluxes observed at a few points within some ponds. Flux rates were approximately three times higher than those reported for the wet phase of similarly sized permanent ponds (Holgerson and Raymond 2016). Seasonality strongly influenced the magnitude of emissions, with the highest fluxes observed in summer. This seasonal effect also shaped the influence of hydroperiod, which was significant only during summer and interacted with climate, leading to higher emissions in Mediterranean ponds with extended hydroperiods. The overall pattern was largely driven by the interaction between sediment temperature and water content, which followed a curvilinear relationship with an apparent threshold effect. Other local factors, such as pond conservation status and carbonate content, also played a role in modulating $CO_2$ fluxes. Taken together, these findings suggest that changes in temperature, precipitation, hydroperiod, and pond ecological condition, driven by climate change, land use or other human-induced alterations, can significantly affect $CO_2$ emissions from temporary ponds.

### 4.1 $CO_2$ fluxes across ponds: influence of climate, seasonality and hydroperiod

Our study showed high variability of $CO_2$ fluxes both among and within ponds, highlighting the importance of accounting for microhabitats in assessments. As hypothesized (Hypothesis 1), we found significant seasonal variation, emphasizing the importance of the timing of the drying period, with the highest fluxes occurring under summer conditions. However, contrary to our expectations, overall $CO_2$ emissions did not differ significantly between climate regions (Mediterranean vs Temperate). This finding aligns with that of Keller et al. (2020) and suggests that local factors may outweigh broader climatic influences. The $CO_2$ dry fluxes reported by Keller et al. (2020) (mean ± SD = 3204 ± 2652 mg C m$^{-2}$ d$^{-1}$ or 267 ± 221 mmol m$^{-2}$ d$^{-1}$, $N = 27$), measured exclusively during the summer, were significantly higher (t-test $p = .006$) than the average fluxes obtained in our study (mean ± SD = 1398 ± 1201 mg C m$^{-2}$ d$^{-1}$ or 117 ± 100 mmol m$^{-2}$ d$^{-1}$, $N = 30$), which encompassed a broader seasonal range including both summer and autumn (Table 4). In contrast, there was no significant difference when comparing only summer emissions (t-test $p = .15$). Similarly, our emission values fall within the range reported for other temporary ponds that captured different stages of the drying period (i.e., seasonal variation) (Catalán et al., 2014; Obrador et al., 2018) (Table 4). This finding underscores the importance of assessing $CO_2$ emissions throughout the entire dry period, as fluxes can vary substantially over time and between seasons.

**Table 4. $CO_2$ emissions from ponds during dry and wet phases, including mean values and standard deviation (SD) from this study and previously published studies conducted across different climate regions, seasons, and hydrological conditions.**

| Study | Location | Climate | $CO_2$ emissions (mg C m$^{-2}$ d$^{-1}$) | SD | Season | Pond type |
|---|---|---|---|---|---|---|
| **Our study** | | | | | | |
| *Overall Ponds* | Europe | Temperate, Mediterranean | 1398 | 1201 | Summer, autumn | Temporary, semi-permanent, permanent |
| *Ponds in summer* | Europe | Temperate, Mediterranean | 2111 | 1739 | Summer | Temporary, semi-permanent, permanent |
| *Ponds in autumn* | Europe | Temperate, Mediterranean | 598 | 464 | Autumn | Temporary, semi-permanent, permanent |
| **Inundated** | | | | | | |
| *Holgerson and Raymond (2016)* | Global (< 0.001 km$^2$) | Most general climates included | 422 | 63 | Not specified | Permanent |
| *Holgerson and Raymond (2016)* | Global (0.001-0.001 km$^2$) | Most general climates included | 255 | 71 | Not specified | Permanent |
| *DelVecchia et al. (2019)* | USA | Subalpine | 129 | 123 | Summer | Temporary, semi-permanent, permanent |
| **Dry** | | | | | | |
| *Keller et al. (2020)* | Global | Tropical, temperate, polar | 3204 | 2652 | Summer | Temporary |
| *Catalán et al. (2014)* | Spain | Mediterranean | 1456 | 1657 | Summer, autumn | Temporary |
| *Obrador et al. (2018)* | Spain | Mediterranean | 1294 | 1247 | Spring, summer, autumn | Temporary |
| *DelVecchia et al. (2019)* | USA | Subalpine | 1813 | 321 | Summer | Temporary, semi-permanent and permanent |
| *Martinsen et al. (2019)* | Sweden | Temperate | 2995 | 2090 | Summer | Not specified |
| *Fromin et al. (2010)* | France | Mediterranean | 120-7200 | n.d. | All seasons | Temporary |
| *DelVecchia et al. (2021)* | USA | Alpine | 3978 | 138 | Summer | Temporary, semi-permanent and permanent |
| *DelVecchia et al. (2021)* | USA | Subalpine | 1714 | 541 | Summer | Temporary, semi-permanent and permanent |

This temporal variability was also reflected in the role of hydroperiod length, which showed a clear seasonal dependence modulated by regional climate. Significant differences in emissions were observed only during summer, with contrasting trends across climate regions. Notably, the hypothesized effect of hydroperiod (Hypothesis 2) was supported only in Mediterranean ponds, where longer hydroperiods were associated with higher $CO_2$ emissions. Hydroperiod showed a moderate positive correlation with sediment water content (r = .47; Fig. B1), suggesting that longer hydroperiods are associated with higher sediment water content. This relationship could partially explain the effect of hydroperiod on $CO_2$ emissions. However, other environmental factors are likely to contribute as well. Moreover, hydroperiod length was positively correlated with increased tryptophan-like fluorescence (C3), indicating the accumulation of less processed humic-

like (C2) and more labile organic matter in the sediments (Fig. S4). The effect of both humic-like and tryptophan-like components on $CO_2$ emissions exhibited clear seasonal dependence, with significant impacts observed during summer (Fig. S5). Hydroperiod duration influences organic matter accumulation during the wet phase, which may lead to increased $CO_2$ emissions as sediments become more aerobic during the dry season (Downing, 2010; Fromin et al., 2010; Jarvis et al., 2007; Marcé et al., 2019; Obrador et al., 2018; Rulík et al., 2023). In the Mediterranean, temporary ponds are experiencing shorter hydroperiods, becoming increasingly intermittent or disappearing altogether (Gómez-Rodríguez et al., 2010). Similarly, ponds in the Temperate region are vulnerable to shifts from permanent to semi-permanent or temporary states (Čížková et al., 2013; Lee et al., 2015). These transitions toward more temporary systems may have important climate-warming implications, as exposed sediments can amplify $CO_2$ emissions (Marcé et al., 2019). However, these emissions are strongly influenced by the timing of pond drying, with higher magnitudes occurring under summer conditions. Our results in the Mediterranean region indicate that climate-driven transitions toward shorter hydroperiods and more temporary pond conditions may actually reduce $CO_2$ fluxes during the dry phase. This finding highlights the need to evaluate how shifts in hydroperiods could impact $CO_2$ emissions across regions; however, a complete understanding would require integrating both wet and dry phases.

### 4.2 Drivers of $CO_2$ fluxes during the dry phase

Our study identified the key drivers explaining $CO_2$ emissions from dry pond sediments as the interaction between sediment temperature and water content, conservation status, and carbonate content. As hypothesized (Hypothesis 3), local factors, particularly the temperature-water content interaction, played a crucial role in $CO_2$ emissions from dry sediments, aligning with previous studies on dry flux emission (Keller et al., 2020; Marcé et al., 2019; Martinsen et al., 2019; Obrador et al., 2018; Oertel et al., 2016). Our results revealed a non-linear interaction between sediment water content and temperature, where water content exerts a threshold effect modulated by temperature, constraining $CO_2$ emissions at both low and high extremes of these variables. Low sediment water content limits microbial activity, reducing $CO_2$ fluxes, while saturated conditions restrict oxygen availability and gas diffusion, suppressing aerobic respiration (Almeida et al., 2019; Oertel et al., 2016; Sánchez-Carrillo, 2009). This effect is temperature-dependent, as rising temperatures stimulate microbial metabolism and respiration, directly increasing $CO_2$ emissions (Fromin et al., 2010; Gilbert et al., 2017). Water content also influences sediment temperature, further affecting $CO_2$ fluxes (Sabater et al., 2016; Sponseller, 2007). These results highlight that sediment temperature and water content are interdependent drivers of carbon fluxes, underscoring the complex interplay between abiotic and biotic factors regulating these systems (Marcé et al., 2019).

In our study, dry sediment temperatures varied widely, ranging from 7 to 47.5 °C (mean ± SD: 18.8 ± 7.8 °C), with significant seasonal differences. In contrast, sediment water content did not differ significantly between seasons, although it showed a broad range from 0.89% to 94.2% (mean ± SD: 34.8 ± 24%). Our analysis predicts that $CO_2$ emissions peak at sediment temperatures around 27.7 ºC and sediment water content levels between 27% to 44%. These results indicate that maximum $CO_2$ fluxes occur under higher sediment temperature and moderate water content conditions, likely optimizing

microbial metabolism. These findings align with Pozzo-Pirotta et al. (2022) and Gómez-Gener et al. (2016), who reported the highest emissions at temperature ranges from 18 to 25.5 °C. Conversely, temperatures exceeding ~30 °C may promote

evaporation, reduce water availability, and ultimately lower respiration rates due to enzyme inhibition and physiological stress (Lellei-Kovács et al., 2011). The temperature-water content relationship explains the observed seasonal pattern, with elevated temperatures enhancing microbial activity and emissions during summer, and reduced activity during autumn. This pattern can also partly explain the variation in emissions across ponds with different hydroperiod lengths. Furthermore, the absence of significant differences in $CO_2$ fluxes between the two climate regions may reflect the predominance of local-scale

drivers, as both sediment temperature and water content conditions favourable for microbial respiration, and consequently $CO_2$ emissions, were present in ponds across both Mediterranean and Temperate climate regions.

We also found a negative relationship between $CO_2$ fluxes and carbonate content, suggesting that carbonate-rich sediments may reduce mineralization rates, either by limiting microbial activity or through abiotic processes such as carbonate reactions. Although biological activity is the primary source of $CO_2$ fluxes from dry sediments, abiotic processes, such as

pedochemical and geological reactions (e.g., carbonate dissolution), may also contribute, albeit typically to a lesser extent (Rey, 2015). This relationship may arise because, in some regions, carbonaceous substrates modulate $CO_2$ levels during dry periods of low biological activity through weathering and precipitation reactions involving inorganic carbon (Roland et al., 2013; Hanken, Bjørlykke and Nielsen, 2015). This could explain the occasional negative $CO_2$ fluxes observed in our study (mean $\pm$ SD: - 257.6 $\pm$ 191.3 mg C m$^{-2}$ d$^{-1}$), which did not differ between seasons (t-test, $p$ = .79). These values are within the

range reported in studies using closed chambers, including Keller et al. (2019) ( -324 mg C m$^{-2}$ d$^{-1}$), Ma et al. (2013) ( -290 and -436 mg C m$^{-2}$ d$^{-1}$ for under-canopy and inter-plant spaces, respectively), and Koschorreck et al. (2022) (-1,440 to 13,620 mg C m$^{-2}$ d$^{-1}$), with a similar proportion of negative fluxes (6%). Since all our measurements were conducted in reflective chambers that excluded light, it is unlikely that these negative fluxes resulted from photosynthetic $CO_2$ uptake by residual phytoplankton macrophytes or cryptobiotic crusts. Instead, they likely reflect sediment physico-chemical processes

associated with inorganic reactions (Ma et al., 2013; Marcé et al., 2019).

Moreover, as hypothesized (Hypothesis 4), ponds in better ecological condition, i.e., those with higher conservation status values, exhibited higher $CO_2$ emissions during the dry phase. Well-conserved ponds tend to support higher macrophyte diversity and vegetation cover, which enhances the accumulation and stabilization of organic carbon in sediments, promoting carbon sink during wet phases (Akasaka et al., 2010; Garcia-Murillo et al., 2025; Nag et al., 2023). In our study, pond

conservation status (ECELS index) was positively correlated with both macrophyte coverage and PVI ($p$ < .001). In turn, macrophyte cover correlated positively with sediment organic matter and DOC ($p$ < .001). During dry phases, the exposure of previously vegetated sediments to air can trigger plant senescence and increase microbial activity, leading to elevated $CO_2$ emissions (Catalán et al., 2014; Martinsen et al., 2019). This mechanism may help explain the counterintuitive relationship between higher conservation status and increased $CO_2$ emissions during the dry phase. Nonetheless, these higher emissions

may be offset during the wet phase by lower emissions, driven by enhanced primary production and greater carbon burial (Page and Dalal, 2011; Morant et al., 2020; Zou et al., 2022). For example, Taylor et al. (2019) found that carbon burial rates

ranged between 216 and 676 mg C $m^{-2}$ $d^{-1}$ depending on ponds' vegetation. Sharma et al. (under revision) reported that $CO_2$ fluxes in wetland bare zones averaged $1555 \pm 3082$ mg C $m^{-2}$ $d^{-1}$, with no significant differences between light and dark conditions. In contrast, fluxes in wetlands vegetated zones were much more variable, reaching $4234 \pm 5789$ mg C $m^{-2}$ $d^{-1}$

under dark conditions and $-1066 \pm 3715$ mg C $m^{-2}$ $d^{-1}$ under light conditions. In line with observations by Madaschi et al. (2025), vegetated soils exhibited greater ecosystem respiration than bare soils; however, their net ecosystem exchange (NEE) was negative, indicating a net $CO_2$ uptake driven by enhanced gross primary production (GPP). Reported NEE values were $1740 \pm 324$ mg C $m^{-2}$ $d^{-1}$ for bare soils and $-912 \pm 288$ mg C $m^{-2}$ $d^{-1}$ for vegetated soils. Our $CO_2$ flux measurements were taken from bare sediments, excluding vegetated areas, and therefore do not reflect the influence of primary production or net

ecosystem metabolism. Consequently, the overall $CO_2$ fluxes from these systems are probably lower than those reported here. Finally, conservation status, represented by the ECELS index, which integrates pond morphology, human impacts, vegetation abundance, and water quality, appears to be a stronger predictor of $CO_2$ emissions during the dry phase than individual factors such as land use or trophic status proxies (e.g., chlorophyll a, TN, TP). This highlights the value of integrative ecological indices over isolated proxies when evaluating pond function.

**4.3 Implications of our study in the context of global change**

Overall, our results emphasize the importance of assessing $CO_2$ emissions throughout the entire dry period, as key drivers such as sediment temperature and water content can fluctuate over short time scales and are strongly influenced by climatic conditions. Capturing this temporal variability is critical, as single-time-point measurements may underestimate the actual emissions dynamics of these ecosystems. Incorporating this variability into carbon budgets will improve their accuracy and

enhance the design of climate mitigation strategies that consider $CO_2$ emissions from dry sediments. Despite their relevance, studies quantifying $CO_2$ fluxes during dry phases remain scarce, creating a critical knowledge gap that limits the inclusion of temporary systems into global GHG inventories (Keller et al., 2020; Lauerwald et al., 2023; Marcé et al., 2019).

Our findings also suggest that projected increases in temperature are likely to elevate sediment temperatures, which, especially when combined with episodic rewetting events, may substantially enhance $CO_2$ emissions during dry phases. In

addition, ponds with longer hydroperiods exhibited higher dry-phase $CO_2$ fluxes than ephemeral systems in Mediterranean regions. However, to fully understand how shifts in hydroperiod toward increased intermittency affect carbon dynamics, future studies should integrate measurements from both wet and dry phases. Understanding these processes is essential in the context of climate change, as climate regions are facing divergent hydrologic trends (Pekel et al., 2016; Wang et al., 2018; Vicente-Serrano et al., 2022; Bevacqua et al., 2024). Given that extreme events, such as the severe 2022 European drought

(Copernicus Climate Change Service (C3S), 2023), are expected to become more frequent, and that anthropogenic pressures continue to intensify, inland water systems are undergoing rapid transformations that will increasingly influence the global carbon cycle.

**Conclusion**

Our results highlight the need to integrate $CO_2$ emissions across all stages of the dry season to achieve accurate estimates of
fluxes in ponds. Although no significant differences in emissions were observed among climatic regions, key drivers such as hydroperiod length, sediment temperature, and sediment water content are inherently linked to climate. Moreover, ponds with better conservation status emitted more $CO_2$ during the dry phase; however, a comprehensive integration with emissions from the wet phase is still required. Understanding these dynamics is crucial for predicting carbon fluxes in pond ecosystems under future climate and land-use scenarios.

## 530 Appendix A: Pond sampling, hydro-geomorphological and sediment characteristics

**Table A1. Geographic location and sampling details of the studied ponds, including pond identifiers, pondscape name, hydrological classification, coordinates (longitude and latitude), and sampling dates and times during summer and autumn campaigns.**

| Pond ID | Pond Code | Country | Pondscape | Hydrological classification | Longitude | Latitude | Date Summer | Time summer start | Time summer finish | Date Autumn | Time autumn starts | Time autumn finish |
|---|---|---|---|---|---|---|---|---|---|---|---|---|
| SP008 | ALB_5 | Spain | Albera | Temporary | 2.980 | 42.397 | 6/7/2022 | 13:47:00 | 14:47:00 | 14/11/2022 | 15:17:00 | 16:17:00 |
| SP014 | ALB_3 | Spain | Albera | Temporary | 2.959 | 42.379 | 6/7/2022 | 11:37:00 | 12:37:00 | 14/11/2022 | 12:40:00 | 13:50:00 |
| SP019 | ALB_4 | Spain | Albera | Temporary | 2.983 | 42.382 | 6/7/2022 | 16:16:00 | 17:31:00 | 14/11/2022 | 10:00:00 | 11:20:00 |
| SP026 | OSO_1 | Spain | Osona | Temporary | 2.352 | 41.961 | 7/7/2022 | 08:07:00 | 09:08:00 | 15/11/2022 | 09:42:00 | 10:42:00 |
| SP028 | OSO_3 | Spain | Osona | Semi-permanent | 2.363 | 41.964 | 7/7/2022 | 09:46:00 | 10:46:00 | 15/11/2022 | 16:02:00 | 16:37:00 |
| SP029 | OSO_4 | Spain | Osona | Semi-permanent | 2.374 | 41.968 | 7/7/2022 | 14:29:00 | 15:29:00 | 15/11/2022 | 11:52:00 | 12:52:00 |
| SP030 | OSO_5 | Spain | Osona | Temporary | 2.375 | 41.963 | 7/7/2022 | 13:07:00 | 14:07:00 | 15/11/2022 | 13:36:00 | 14:31:00 |
| SP032 | SEL_1 | Spain | Selva | Temporary | 2.718 | 41.829 | 4/7/2022 | 11:26:00 | 13:06:00 | 24/11/2022 | 09:42:00 | 11:20:00 |
| SP035 | SEL_4 | Spain | Selva | Temporary | 2.725 | 41.819 | 4/7/2022 | 14:54:00 | 15:54:00 | 24/11/2022 | 12:31:00 | 13:10:00 |
| SP040 | GAR_3 | Spain | Garrtoxa | Semi-permanent | 2.501 | 42.284 | 13/7/2022 | 12:15:00 | 13:15:00 | - | - | - |
| SP041 | GAR_4 | Spain | Garrtoxa | Temporary | 2.498 | 42.289 | 13/7/2022 | 09:33:00 | 10:33:00 | - | - | - |
| SP043 | GAR_6 | Spain | Garrtoxa | Temporary | 2.556 | 42.268 | 13/7/2022 | 15:25:00 | 16:13:00 | - | - | - |
| SP044 | GAV_1 | Spain | Gavarres | Temporary | 2.922 | 41.836 | 5/7/2022 | 16:09:00 | 17:09:00 | 16/11/2022 | 15:10:00 | 14:10:00 |
| SP045 | GAV_2 | Spain | Gavarres | Semi-permanent | 2.878 | 41.842 | 12/7/2022 | 10:55:00 | 11:55:00 | - | - | - |
| SP046 | GAV_3 | Spain | Gavarres | Temporary | 2.895 | 41.870 | 5/7/2022 | 11:34:00 | 12:34:00 | 23/11/2022 | 09:53:00 | 10:53:00 |
| SP049 | GAV_6 | Spain | Gavarres | Temporary | 2.895 | 41.865 | 12/7/2022 | 12:50:00 | 13:50:00 | 23/11/2022 | 11:40:00 | 12:40:00 |
| BE059 | FB1_EX2 | Belgium | Flemish_Brabant | Permanent | 4.943 | 50.929 | - | - | - | 5/10/2022 | 18:49:00 | 19:36:26 |
| BE065 | LB2_IN2 | Belgium | Bocholt | Temporary | 5.605 | 51.195 | - | - | - | 5/10/2022 | 15:00:00 | 15:45:00 |
| DE001 | MUN_56 | Germany | Muncheberg | Permanent | 14.112 | 52.474 | - | - | - | 29/9/2022 | 12:43:00 | 14:12:00 |
| DE007 | SCH_07 | Germany | Schoneiche | Temporary | 13.699 | 52.481 | 11/7/2022 | 13:46:00 | 17:05:00 | - | - | - |
| DE011 | LIE_12 | Germany | Lietzen | Temporary | 14.353 | 52.459 | 12/8/2022 | 11:30:00 | 12:33:00 | - | - | - |
| DE012 | LIE_09 | Germany | Lietzen | Temporary | 14.353 | 52.460 | 15/7/2022 | 12:09:00 | 16:05:00 | - | - | - |
| DE016 | QUI_ex5 | Germany | Quillow | Semi-permanent | 13.623 | 53.322 | 28/9/2022 | 14:08:00 | 15:07:00 | - | - | - |
| DE019 | QUI_ex2 | Germany | Quillow | Temporary | 13.567 | 53.307 | 14/7/2022 | 10:34:00 | 11:38:00 | 5/10/2022 | 09:10:00 | 11:00:00 |
| DE028 | MUN_41 | Germany | Muncheberg | Semi-permanent | 14.145 | 52.472 | 5/8/2022 | 13:24:00 | 17:09:00 | 6/10/2022 | 11:04:00 | 12:30:00 |
| DE030 | MUN_105 | Germany | Muncheberg | Temporary | 14.145 | 52.522 | 12/7/2022 | 14:54:00 | 16:20:00 | 30/9/2022 | 11:06:00 | 13:13:00 |
| DE031 | MUN_201 | Germany | Muncheberg | Temporary | 14.133 | 52.530 | - | - | - | 30/9/2022 | 14:58:00 | 15:09:00 |
| DK018 | PAE6 | Denmark | Aero | Temporary | 10.319 | 54.902 | 11/7/2022 | 11:39:15 | 12:41:46 | 15/11/2022 | 11:53:45 | 12:01:15 |
| DK042 | PAE2 | Denmark | Aero | Permanent | 10.348 | 54.866 | - | - | - | 14/11/2022 | 11:06:00 | 11:58:00 |
| DK034 | PAE4 | Denmark | Aero | Semi-permanent | 10.329 | 54.874 | - | - | - | 15/11/2022 | 07:56:00 | 09:41:02 |

**Table A2. Hydro-geomorphological and sediment characteristics of the studied ponds.**

| Pond | Country | Climate | Area (m²) | Maximum depth (cm) | Hydroperiod length (months) | Season | Temperature (°C) | Water content (%) | pH | Conductivity (μS cm⁻¹) | Carbonate content (%) | Organic matter (%) | DOC (mg C g⁻¹) |
|---|---|---|---|---|---|---|---|---|---|---|---|---|---|
| SP008 | Spain | Mediterranean | 2555.36 | 127 | 4 | Summer | 24.94 | 12.80 | 4.99 | 314.6 | 0.73 | 16.01 | 2.36 |
| | | | | | | Autumn | 18.2 | 19.51 | 4.93 | 420.4 | 1.25 | 12.52 | 1.17 |
| SP014 | Spain | Mediterranean | 13465.72 | 121 | 4 | Summer | 22.27 | 29.87 | 4.91 | 595.71 | 1.44 | 23.69 | 1.18 |
| | | | | | | Autumn | 16.44 | 19.07 | 4.77 | 786 | 1.76 | 23.77 | 1.77 |
| SP019 | Spain | Mediterranean | 11455.29 | 125 | 3 | Summer | 25.4 | 19.40 | 5.08 | 389.22 | 0.94 | 17.08 | 1.66 |
| | | | | | | Autumn | 14.66 | 14.32 | 5.05 | 456.22 | 1.34 | 17.68 | 1.52 |
| SP026 | Spain | Mediterranean | 176.62 | 77 | 7 | Summer | 20.15 | 3.59 | 7.53 | 957.5 | 0.15 | 2.78 | 0.39 |
| | | | | | | Autumn | 10.43 | 27.95 | 7.47 | 459.67 | 22.53 | 4.33 | 0.4 |
| SP028 | Spain | Mediterranean | 324.84 | 104 | 12 | Summer | 18.83 | 32.39 | 7.2 | 718.17 | 0.22 | 5.85 | 0.48 |
| | | | | | | Autumn | 12.25 | 41.30 | 7.2 | 318 | 19.72 | 8.74 | 0.62 |
| SP029 | Spain | Mediterranean | 128.52 | 140 | 12 | Summer | 26.74 | 39.30 | 7.24 | 422.6 | 3.52 | 16.78 | 0.56 |
| | | | | | | Autumn | 10.98 | 30.93 | 7.38 | 334.8 | 18.95 | 5.21 | 0.48 |
| SP030 | Spain | Mediterranean | 248.35 | 137 | 10 | Summer | 25.83 | 38.78 | 7.22 | 556.33 | 0.21 | 5.39 | 0.74 |
| | | | | | | Autumn | 11.42 | 38.39 | 7.26 | 325.83 | 18.39 | 5.7 | 0.56 |
| SP032 | Spain | Mediterranean | 565.29 | 146 | 2 | Summer | 25.98 | 31.87 | 7.3 | 713.22 | 0.52 | 6.23 | 0.38 |
| | | | | | | Autumn | 9.29 | 24.47 | 7.52 | 336.22 | 1.46 | 6.26 | 0.47 |
| SP035 | Spain | Mediterranean | 307.70 | 118 | 12 | Summer | 27.82 | 38.32 | 6.97 | 244.34 | 0.1 | 6.17 | 0.63 |
| | | | | | | Autumn | 12.68 | 29.42 | 6.84 | 134.88 | 0.69 | 4.4 | 0.5 |
| SP040 | Spain | Mediterranean | 86.05 | 85 | 10 | Summer | 26.98 | 43.65 | 7.39 | 480.25 | 20.91 | 7.15 | 1.32 |
| SP041 | Spain | Mediterranean | 84.39 | 72 | 4 | Summer | 27.36 | 2.51 | 7.76 | 282.8 | 5.31 | 10.37 | 1.47 |
| SP043 | Spain | Mediterranean | 157.48 | 58 | 8 | Summer | 39.33 | 2.15 | 7.67 | 462.5 | 21.84 | 14.33 | 1.92 |
| SP044 | Spain | Mediterranean | 1205.68 | 93 | 5 | Summer | 33.17 | 3.63 | 7.39 | 628.33 | 0.89 | 11.71 | 0.83 |
| | | | | | | Autumn | 15.03 | 33.91 | 7.84 | 245.83 | 8.52 | 12.37 | 1.23 |
| SP045 | Spain | Mediterranean | 65.42 | 148 | 7 | Summer | 26.77 | 13.49 | 6.38 | 131.33 | 0.44 | 1.78 | 0.33 |
| SP046 | Spain | Mediterranean | 504.42 | 72 | 3 | Summer | 32.76 | 5.02 | 5.61 | 106.64 | 0.13 | 15.28 | 4.6 |
| | | | | | | Autumn | 8.56 | 20.16 | 5.61 | 106.64 | 0.8 | 3.99 | 0.63 |
| SP049 | Spain | Mediterranean | 99.40 | 73 | 6 | Summer | 32 | 3.97 | 6.11 | 147.22 | 1.3 | 7.69 | 0.46 |
| | | | | | | Autumn | 9.08 | 20.56 | 5.87 | 96.52 | 0.83 | 2.27 | 0.29 |
| BE059 | Belgium | Temperate | 98 | 87 | 12 | Autumn | 16.33 | 21.04 | 6.11 | 485.53 | 0.73 | 5.08 | 0.64 |
| BE065 | Belgium | Temperate | 380 | 83.75 | 6 | Autumn | 16.55 | 9.31 | 6.17 | 59.83 | 0.33 | 1.15 | 0.3 |
| DE001 | Germany | Temperate | 2250 | 262 | 12 | Autumn | 12.3 | 80.96 | 7.25 | 453.67 | 1.54 | 46.89 | 7.67 |
| DE007 | Germany | Temperate | 1350 | 170 | 8 | Summer | 19.99 | 63.59 | 6.73 | 305.33 | 1.08 | 32.2 | 2.04 |
| DE011 | Germany | Temperate | 1500 | 83 | 9 | Summer | 21.5 | 66.37 | 7.35 | 457.83 | 14.8 | 20.75 | 2.33 |
| DE012 | Germany | Temperate | 2800 | 69 | 8 | Summer | 19.95 | 68.92 | 7.57 | 1484.5 | 13.41 | 21.93 | 2.15 |
| DE016 | Germany | Temperate | 150 | 160 | 12 | Autumn | 13.8 | 61.11 | 6.96 | 462.2 | 2.12 | 11.32 | 1.8 |
| DE019 | Germany | Temperate | 60 | 70 | 9 | Summer | 19.7 | 41.15 | 5.61 | 615 | 2.28 | 12.53 | 1.28 |
| DE028 | Germany | Temperate | 600 | 115 | 10 | Summer | 24.64 | 23.06 | 6.65 | 741 | 1.59 | 8.16 | 0.79 |
| | | | | | | Autumn | 13.52 | 19.46 | 6.29 | 1504.8 | 1.39 | 7.26 | 1.52 |
| DE030 | Germany | Temperate | 1200 | 55 | 10 | Summer | 23.81 | 75.01 | 7.12 | 581.2 | 2.04 | 33.58 | 1.89 |
| | | | | | | Autumn | 10.83 | 79.48 | 6.77 | 450.83 | 3.7 | 39.69 | 2.84 |
| DE031 | Germany | Temperate | 800 | 45 | 7 | Autumn | 11.89 | 76.53 | 7.18 | 1087.6 | 5.07 | 69.51 | 4.04 |
| DK018 | Denmark | Temperate | 820 | 78 | 4 | Summer | 20.48 | 51.13 | 6.75 | 302.83 | 1.08 | 13.27 | 1.5 |
| | | | | | | Autumn | 8.9 | 55.58 | 6.88 | 107.6 | 1.02 | 12.73 | 1.44 |
| DK034 | Denmark | Temperate | 190 | 103 | 10 | Autumn | 7.7 | 71.39 | 7.54 | 507.02 | 21.39 | 14.59 | 3.48 |
| DK042 | Denmark | Temperate | 645 | 100 | 12 | Autumn | 9.42 | 80.25 | 6.54 | 529.99 | 1.87 | 29.63 | 9.48 |

**Table A3. List of variables measured in this study, grouped by spatial scale: regional, landscape, local, sediment and water characteristics, dissolved organic matter and components of organic matter based on PARAFAC (Parallel Factor Analysis).**

| | Variable | Type | Unit |
|---|---|---|---|
| **Regional** | Temperature 40-year average (1978-2018) | Continuous | K |
| | Precipitation 40-year average (1978-2018) | Continuous | mm s$^{-1}$ |
| | Climate | Categorical | (Mediterranean, Temperate) |
| | Annual temperature (2022) | Continuous | °C |
| | Annual precipitation (2022) | Continuous | mm |
| | Latitude | Continuous | Decimal degrees (WGS 84) |
| **Landscape** | Open nature (5 m and 100) | Continuous | % |
| | Forest (5 m and 100) | Continuous | % |
| | Pasture (5 m and 100) | Continuous | % |
| | Arable (5 m and 100) | Continuous | % |
| | Grassland (5 m and 100) | Continuous | % |
| | Urban (5 m and 100) | Continuous | % |
| **Local pond characteristics** | ECELS index (Conservation status) | Continuous | 0-100 |
| | Coverage | Continuous | % |
| | PVI (plant volume inhabited) | Continuous | % |
| | Hydroperiod length | Continuous | months |
| | Area | Continuous | m$^2$ |
| | Maximum depth | Continuous | cm |
| **Water characteristics** | Chlorophyll a | Continuous | µg L$^{-1}$ |
| | Total Nitrogen | Continuous | mg L$^{-1}$ |
| | Total phosphorus | Continuous | mg L$^{-1}$ |
| | Dissolved organic carbon (DOC) | Continuous | mg L$^{-1}$ |
| **Sediment characteristics** | Temperature | Continuous | °C |
| | Water content | Continuous | % |
| | Organic matter | Continuous | % |
| | Carbonate content | Continuous | % |
| | pH | Continuous | - |
| | Conductivity | Continuous | µS cm$^{-1}$ |
| | Texture | Categories | (Clay, Sandy, Loamy) |
| **Dissolved organic matter** | Dissolved organic carbon (DOC) | Continuous | mg C g$^{-1}$ |
| | Absorbance at 254 nm (Abs254) | Continuous | - |
| | Absorbance at 300 nm (Abs300) | Continuous | - |
| | BIX (biological index) | Continuous | - |
| | HIX (Humification index) | Continuous | - |
| | FI (fluorescence index) | Continuous | - |
| | SUVA (Specific Ultraviolet Absorbance) | Continuous | L mg C$^{-1}$ m$^{-1}$ |
| **PARAFAC** | C1 Terrestrial humic-like | Continuous | - |
| | C2 Humic-like | Continuous | - |
| | C3 Tryptophan-like | Continuous | - |

**Appendix B: Correlation among variables**

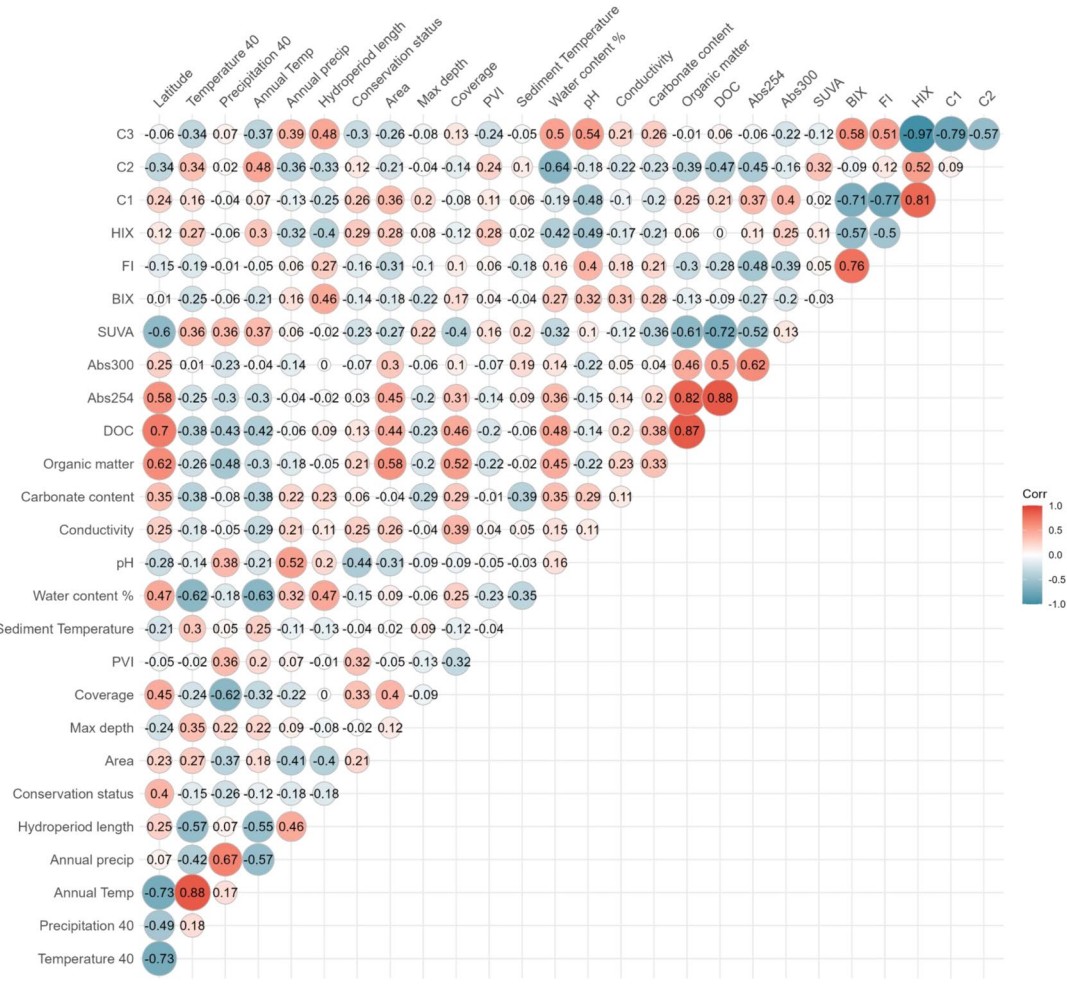


**Figure B1. Correlation matrix of environmental and sediment variables assessed in relation to $CO_2$ fluxes in ponds. Pearson correlation coefficients (R) are displayed within the cells. The colour scale represents the strength and direction of the correlations, with red indicating positive and blue indicating negative relationships. Abbrev. C1 = terrestrial humic-like, C2 = humic-like and C3 = tryptophan-like.**

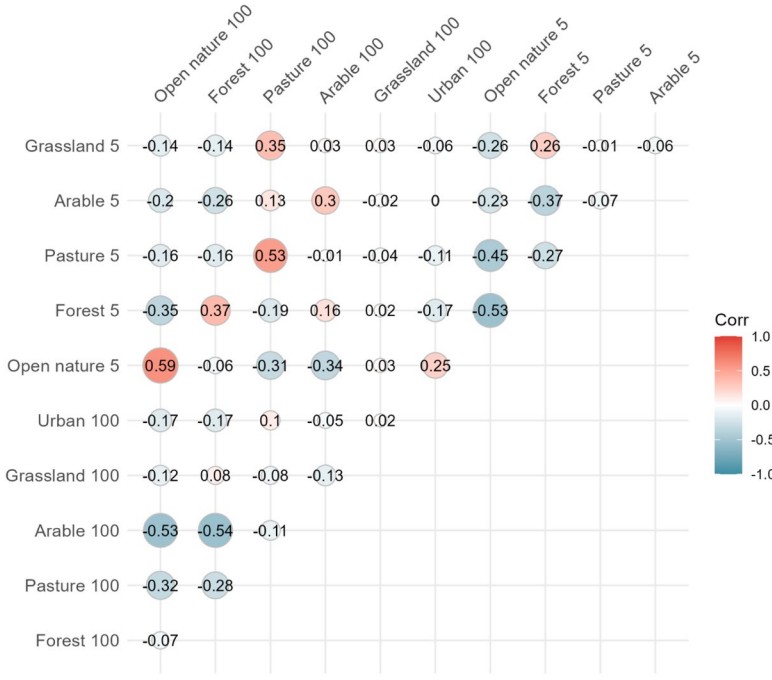


**Figure B2.** Correlation matrix of land use variables at 5m and 100m scales assessed in relation to $CO_2$ fluxes in ponds. Pearson correlation coefficients (R) are shown within the matrix. The colour scale represents correlation strength and direction, with red indicating positive and blue indicating negative relationships.

## Data availability

The datasets used in this study will be available upon reasonable request.

## Author contribution

CT and SB conceived the study and helped supervise the project; VFA and CT developed the theory and performed the computations; VFA, LMP, CT, SB, PL, TB, and TD collected the field data; VFA, CT and LMP performed the laboratory analyses; RM supported the sediment analyses and provided material and infrastructure; TB and TD designed the $CO_2$
sensors and scripts for gas data collection; VFA analysed the data with support from CT, TB, and RM; VFA wrote the manuscript with substantial input from CT and SB; all authors revised, edited and approved the final version of the manuscript.

**Competing interests**

The contact author has declared that none of the authors has any competing interests.

**Acknowledgements**

We thank the PONDERFUL consortium; especially the staff of Aquatic Ecology Group (GEA) of the University of Vic - Central University of Catalonia, the GRECO research teams of the Institute of Aquatic Ecology of the University of Girona, Aarhus University (AU), and the Leibniz-Institute of Freshwater Ecology and Inland Fisheries (IGB), for assistance in the field and technical assistance. The authors acknowledge the use of the AI language model ChatGPT (OpenAI) to assist in improving the English language and writing style of this manuscript.

**Financial support**

This research has received funding from the European Union's research and innovation programme (H2020) under grant agreement No 869296 – The PONDERFUL Project. SB, TAD, TB, and CT have also received funding from the MCIN/AEI/10.13039/501100011033/UE under the Biodiversa + TRANSPONDER grant (PCI2023-145983-2). VFA has a PhD fellowship from AGAUR-FI predoctoral program (2024 FI-3 00755) Joan Oró of the Secretariat of Universities, Research of the Department of Research, Universities of the Generalitat de Catalunya, and the European Social Fund Plus. CT is a CONICET (Argentinean Council of Science) researcher. RM participated through the project Alter-C (Spanish Agencia Estatal de Investigación grant PID2020-114024GB-C32/AEI/10.13039/501100011033).

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
