# Peer review of "Drivers of CO2 emissions during the dry phase in Mediterranean and Temperate ponds"

_EGUsphere, 2025_

## Referee Comment (RC1)

**Review of the manuscript "Drivers of $CO_2$ emissions during the dry phase of Mediterranean and Temperate ponds" by Frutos-Aragón et. al for Biogeosciences:**

The study examines $CO_2$ fluxes of dry sediments from 14 temperate and 16 Mediterranean ponds in Europe during their dry phase in summer and/or autumn of 2022. These fluxes were measured using chambers. Additionally, sediment and water analyses were conducted to characterize the sites and to investigate the differences.

I read the manuscript with great interest and believe that the study is promising, but that some improvements are necessary.

General comments or questions:

- In the abstract, the methodology for measuring $CO_2$ fluxes is missing.
- How exactly and to what extent does your study fill the knowledge gap you mentioned in the introduction?
- In the introduction, a concluding sentence on how the questions will be answered is missing.
- The coordinates of the sampling or measurement points within the sites, the sampling dates, and the names of the studied ponds are missing.
- The chamber description still lacks some information (see the detailed comments).
- For me, it is unclear how often or at what frequency the $CO_2$ measurements were conducted in each season. Were there temporal replicates?
- In the manuscript, there is no information on whether the data requirements for the statistical tests used (e.g., ANOVA or t-test) are met.
- I miss a conclusion chapter.
- The graphics are slightly pixelated, and the image quality could be improved.

Detailed comments and suggestions:

- Abstract: The methodology is missing
- L23ff: What do you mean by hydroperiod?
- L26: c.?
- L37f: It would be good to specify the $CO_2$ emission value of ponds.
- L104f: Instead of spanning latitudes or longitudes, it would be better to add the actual coordinates of some sites and refer to the appendix.
- Figure 1: Since graphs a) and b) are already zoomed in, it would be helpful to include an overview graph with all sites, countries, climate regions, etc.
- It would be interesting to categorize the sites in Figure 1 according to the dry or wet phase definition written in L109 ff.
- Method chapter: Please use the same description style for each instrument. Currently, it differs. You do not have to repeat it if the instrument is already mentioned.
- L117: What do you mean by "dry fluxes"? I also recommend using $N$ instead of n for the number of observations/samples throughout the manuscript.
- L123ff: Why were 40-year averages of annual temperatures and precipitation used instead of the more common 30-year averages for the climatic description of sites?
- L130: You have finally defined what you mean by "hydroperiod length," although the term was already mentioned before. It would have been helpful to define it the first time it was mentioned.
- L157: Why did you use a filter size of 0.7 μm to obtain the dissolved fraction instead of, for example, 0.45 μm?

- L165ff: This subchapter is missing important information about the chamber measurement (e.g., whether it is a through-flow or static system), the chamber type (transparent or opaque), the chamber size (area, height, and volume), and the chamber material. In addition, did you use any additional materials during the measurement, such as tubes or a pump? What time of day did you conduct the chamber measurements at the four to eight spots per pond? How many measurement days were there per season at each pond, and how often were they conducted? Did you correct the measured $CO_2$ concentrations for water vapor?
- L167: I am not familiar with the mentioned sensor. What is its precision, compared to a Licor, Los Gatos, or Picarro gas analyzer, for example? At what frequency does this sensor measure during the five-minute closing time?
- L174f: Explain why one measurement had a different sampling technique and measurement time. Was it a one-hour measurement time or a one-hour closing time?
- L175: What was the size and the material of the vials or syringes? When during the one-hour closing time did you measure, and how much volume did you take? Were your chambers equipped with an overpressure valve?
- L175f: What did you measure with the gas chromatograph? This is unclear to me, as it was never mentioned again in the manuscript.
- L178: Why did the $CO_2$ sensor did not require a calibration?
- L178f: What data did you use for the 3-point average, and why was there background noise?
- L180: Why did you only use the last two to three minutes of each five-minute measurement period? What is the reference for Equation 1?
- L182: Why did you choose to use the carbon unit for the fluxes? Additionally, you used hourly units here, but daily units throughout the manuscript.
- L190: Milli-Q is a brand name, not a water type.
- L191: Why did you use 48 hours?
- L195: Please cite the references that used this as a proxy.
- L220: Table S2 is mentioned before Table S1. Please reconsider the order of the tables.
- L221f: This sentence could be moved to the next subchapter, "Statistical (or data) analysis".
- L224ff: Have you had tested your data for normality?
- L259: The R version is important for repeatability.
- Figure 2: The unit in the y-axis label is missing a bracket. Why does the boxplots have different widths? Does the x-axis label mean Pond ID? To better illustrate the differences and support your results, I would reconsider the representation and categorize by other environmental variables.
- L263: Instead of "overall," I would say "on average" that all your ponds were a $CO_2$ source, as you have also measured a few negative $CO_2$ fluxes.
- L265: It would be better to call them $CO_2$ "fluxes" instead of "emissions" since you have measured some negative values.
- L270: Both are red lines. Please be more precise in the description. What is the mean value of?
- Table 2: I don't think the abbreviation T-FCO$_2$ is necessary or correct here. I would rather use the term "$CO_2$ fluxes" over the four columns on the right, including the unit. Where T-FCO$_2$ and the unit are now, I would put the mean ± SD. I also recommend including the number of observations.
- Figure 3: In the graph, you used n.s., but in the caption, you explained the abbreviation NS. In the supplementary figures, you wrote that an absence indicates no significance. Be consistent.
- L299: Here, you wrote ".01"; in L302, you wrote "0.01" for p. Be consistent throughout the entire manuscript.
- Figure 4: What are the $R^2$ values of the linear regression lines?

- L305: The caption of Figure 4 mentions a dashed line, but I cannot see one in the figure.
- Table 3: I miss the p values you mentioned in L302f.
- Figure 5: I'm not sure what to say about these trend lines. I don't trust them because they look like a point cloud with temperature differences and the influence of the edge effect.
- L345: Sometimes there is a space between the number and "Celsius," and sometimes there isn't. Be consistent throughout the manuscript.
- Table 5: In the text, you always used SD, but in the tables, you used sd. Please be consistent and explain every abbreviation in the table caption. Use the same rounding for all emission values shown here. Since seasonality affects $CO_2$ fluxes in your studied ponds and in most ecosystems outside the tropics, it would be helpful to know what season the reference values were measured.
- L485: The data cannot be reviewed because they are unavailable. I recommend that the authors make the data easily accessible to everyone, not just upon request, for reasons of repeatability and reusability.
- The information about the R packages used is missing and can be added to the references or the methods chapter.
- Why did you use two appendices (A and B)?
- Table A2: Could you add more lines to better separate the categories in the left column and put each variable name in its own row? This would make the table easier to read.
- Figure S1: Variable names, including units, are sometimes split into two lines. This makes the table difficult to read. Please reconsider this.
- Figure S2: In the manuscript, you called it "Pond ID"; here, on the x-axis, it is "Pond Code." Be consistent. The y-axis is missing a unit.
- Figure S3: Please include the number of observations for each box plot.
- Figure S4: There is a typo in the x-axis label: months.
- Figure S5. The sentence about the absence of an asterisk can be removed from the caption.

---

## Author Comment (AC1)

Thank you for taking the time to consider our manuscript. We have reviewed the comments received to address the concerns and your constructive. We have addressed all the comments in the revised manuscript and are grateful for the careful and constructive feedback. The suggestions have substantially improved the clarity and quality of the paper. Below we provide a point-by-point response to each comment, with your comments shown in black and our responses in blue.

**Referee: 1**

**Comments to the Author(s)**

The study examines CO2 fluxes of dry sediments from 14 temperate and 16 Mediterranean ponds in Europe during their dry phase in summer and/or autumn of 2022. These fluxes were measured using chambers. Additionally, sediment and water analyses were conducted to characterize the sites and to investigate the differences.

I read the manuscript with great interest and believe that the study is promising, but that some improvements are necessary.

Reply: Thank you for your time and effort in providing us with constructive feedback to improve our manuscript.

**General comments or questions:**

- In the abstract, the methodology for measuring CO2 fluxes is missing.
  - Reply: We have now included a brief description of the methodology used for measuring CO2 fluxes in the abstract in lines 21-22. We aimed to keep it concise, as abstracts typically require a short overview.
- How exactly and to what extent does your study fill the knowledge gap you mentioned in the introduction?
  - Reply: We acknowledge the reviewer's comment. Our study contributes new data on CO2 emissions from ponds during the dry phase that in comparison with other ecosystems is less represented. Our study highlights the importance of incorporating seasonal frameworks that account for the main local factors controlling CO2 fluxes, such as temperature and water content. Moreover, it introduces hydroperiod length as a first step toward a more integrated understanding of both wet and dry phases, elucidating how transitions from permanent to temporary ponds can affect CO2 dynamics. However, we recognize that this study represents only an initial step, and further research is needed to fully unravel the complexity and variability of these dynamic ecosystems.
- In the introduction, a concluding sentence on how the questions will be answered is missing.

Reply: We have now added a concluding sentence in the introduction to clarify how the research questions are addressed in lines 94-96. We believe that this addition improves the flow and clarity of the section.

"For this reason, our study aims to address this gap by identifying the main drivers of CO2 fluxes during dry periods and examining how the preceding wet phase, in terms of hydroperiod length (i.e., the duration of water presence prior the dry phase in a pond throughout the year) influence them, through a comparison of ponds from contrasting climatic regions across two seasons."

• The coordinates of the sampling or measurement points within the sites, the sampling dates, and the names of the studied ponds are missing.

Reply: We have added a new table in the Appendix (Table A1) that provides the coordinates, the names of the studied ponds, and the corresponding sampling dates (line 526).

• The chamber description still lacks some information (see the detailed comments).

Reply: We added the requested information in the new version of the manuscript (Section 2.3).

• For me, it is unclear how often or at what frequency the CO2 measurements were conducted in each season. Were there temporal replicates?

Reply: We have clarified the frequency of CO2 measurements for each season. Specifically, measurements were conducted once per season in each pond, and no temporal replicates were performed. This clarification has been included to provide a more accurate description of the sampling design (lines 184-185).

"Measurements were conducted once per season at each pond during daytime (08:00–19:00 h), with no temporal replicates within the same season (Table A1; S1)."

• In the manuscript, there is no information on whether the data requirements for the statistical tests used (e.g., ANOVA or t-test) are met.

Reply: We thank the reviewer for this suggestion. As CO2 emissions did not follow a normal distribution in some cases, we revised the analyses using non-parametric Mann–Whitney tests (Wilcoxon rank-sum test; wilcox\_test function, rstatix package; (assambara, 2023). Normality was assessed prior to analysis using Shapiro–Wilk tests (shapiro.test function, stats package). Importantly, the main patterns and differences between groups remain unchanged, while the revised approach provides a more robust statistical framework. We added the information in lines 252-256.

• I miss a conclusion chapter.

Reply: We thank the reviewer for this comment. A conclusion chapter has now been added to the manuscript (lines 521-526).

"Our results highlight the need to integrate CO2 emissions across all stages of the dry season to achieve accurate estimates of fluxes in ponds. Although no significant differences in emissions were observed between climatic regions, key drivers such as hydroperiod length, sediment temperature, and sediment water content are inherently linked to climate. Moreover, ponds with better conservation status emitted more CO2 during the dry phase; however, a comprehensive integration with emissions from the wet phase is still required. Understanding these dynamics is crucial for predicting carbon fluxes in pond ecosystems under future climate and land-use scenarios."

• The graphics are slightly pixelated, and the image quality could be improved.

Reply: The images were prepared following the pixel standards recommended on the journal's webpage. However, we acknowledge that some loss of quality may occur during the PDF upload process.

**Detailed comments and suggestions:**

Abstract: The methodology is missing

Reply: As mentioned above a brief description of the methodology has now been included in the abstract. We kept it concise to maintain the abstract's brevity, while providing enough information to understand the general approach (lines 21-22).

• L23ff: What do you mean by hydroperiod?

Reply: In this study, hydroperiod refers to the duration of time each pond retained water prior to the dry phase, quantified as the number of months with water surface during the 12-month period preceding the last autumn sampling (conducted between late September and November). We have added a brief description in the introduction to clarify this term (lines 94-95).

"Hydroperiod length (i.e., the duration of water presence prior the dry phase in a pond throughout the year)"

• L26: c.?

Reply: The abbreviation "c." stands for "circa", meaning "approximately". To improve clarity, we have replaced it with approximately 27 °C in the manuscript (line 27).

• L37f: It would be good to specify the CO2 emission value of ponds.

Reply: We have now added some representative CO2 emission values from the literature as a reference in lines 40-44.

"However, reported CO2 emissions from ponds are highly variable, ranging from hundreds to several thousand mg C m-2 d-1. For instance, ponds of similar size (<0.001 and 0.001–0.01 km2) reported by Holgerson and Raymond, (2016) emitted on average 254 and 422 mg C m-2 d-1, respectively, whereas exposed pond sediments reported by Keller et al. (2020) range from -73 to 11765 mg C m-2 d-1."

• L104f: Instead of spanning latitudes or longitudes, it would be better to add the actual coordinates of some sites and refer to the appendix.

Reply: The coordinates of the sites have now been included in Table A1 in the Appendix, as recommended.

• Figure 1: Since graphs a) and b) are already zoomed in, it would be helpful to include an overview graph with all sites, countries, climate regions, etc.

Reply: We thank the reviewer for this suggestion. An overview graph including all sites and countries, providing a better zoomed-out view, has been added to the revised manuscript.

Figure 1: Geographic locations of the studied ponds. Left panel: map created using Stadia Maps outdoors basemap and OpenStreetMap data. Map data © OpenStreetMap contributors, © Stadia Maps (https://stadiamaps.com/). Ponds are highlighted in red. Right panel: map showing ponds categorized by hydroperiod: temporary (dark red), semi-temporary (yellow), and permanent (dark blue). Countries are colour-coded according to their climate regions: Mediterranean (orange) and Temperate (light blue).

• It would be interesting to categorize the sites in Figure 1 according to the dry or wet phase definition written in L109 ff.

Reply: The sites in Figure 1 have now been categorized according to the categorical definition described in L109 ff, as suggested. Indicated in the figure attached above.

Method chapter: Please use the same description style for each instrument.
Currently, it differs. You do not have to repeat it if the instrument is already mentioned.

Reply: The Methods section has been revised to ensure a consistent description style for all instruments, standardizing the information to include model and brand, and avoiding repetition when an instrument had already been mentioned.

• L117: What do you mean by "dry fluxes"? I also recommend using N instead of n for the number of observations/samples throughout the manuscript.

Reply: In the literature, "dry fluxes" refers to CO2 fluxes measured from exposed (non-flooded) sediments after the waterbody is no longer inundated. We have clarified this in the revised manuscript (Line 131).

• L123ff: Why were 40-year averages of annual temperatures and precipitation used instead of the more common 30-year averages for the climatic description of sites?

Reply: We used 40-year averages because the climatic database employed in our study (fifth generation of ECMWF atmospheric reanalysis of the global climate (ERA5) of Copernicus Climate Change Service (C3S) (Wouters, 2021)) provides reference values for the period 1978–2018, which corresponds to 40 years. We considered it appropriate to use the available period rather than artificially truncating the dataset, as the longer time frame reduces the influence of short-term anomalies while still representing contemporary climatic conditions. For year of deviance, we included the year of sampling 2022. So, we can capture the trend and the most accurate of the year.

• L130: You have finally defined what you mean by "hydroperiod length," although the term was already mentioned before. It would have been helpful to define it the first time it was mentioned.

Reply: As mentioned, brief description of "hydroperiod length" (i.e., the duration of water presence in a pond over the course of a year) has now been added prior to its definition in the Methods section, to clarify the term and improve readability (lines 94-95).

• L157: Why did you use a filter size of 0.7 μm to obtain the dissolved fraction instead of, for example, 0.45 μm?

Reply: We used 0.7  $\mu$ m GF/F filters because this pore size efficiently separates the dissolved fraction while retaining phytoplankton and larger particles, and reduces clogging compared to finer 0.45  $\mu$ m filters. Moreover, the use of 0.7  $\mu$ m

GF/F filters has been a historical standard in oceanography and limnology, allowing comparability with previous work and consistency in long-term datasets (DFO, 2015; Wetzel and Likens, 2000).

DFO (2015): Fisheries and Oceans Canada, Reference Manual for Limnological Analyses, Publications.gc.ca collection, Fs94-167.

Wetzel, R. G. and Likens, G. E. (2000): Limnological Analyses, 3rd edn., Springer-Verlag, New York, 429 pp., <a href="https://doi.org/10.1007/978-1-4757-3250-4">https://doi.org/10.1007/978-1-4757-3250-4</a>.

L165ff: This subchapter is missing important information about the chamber measurement (e.g., whether it is a through-flow or static system), the chamber type (transparent or opaque), the chamber size (area, height, and volume), and the chamber material. In addition, did you use any additional materials during the measurement, such as tubes or a pump? What time of day did you conduct the chamber measurements at the four to eight spots per pond? How many measurement days were there per season at each pond, and how often were they conducted? Did you correct the measured CO2 concentrations for water vapor?

Reply: We have added the requested methodological details in the revised manuscript. In response to the questions:

We used static, opaque chambers with a surface area of 0.075 m2 and a total volume of 8 L (diameter 345 mm, height 160 mm). The chambers were made of polypropylene (PP) plastic, and no additional materials such as tubes or pumps were used during the measurements (lines 185-192).

Measurements were conducted during daytime between 08:00 and 19:00. Since our study design did not include temporal replicates, only one measurement campaign was carried out per season in each pond (lines 183-184). Because the chambers are opaque, preventing light from entering, diurnal variations are recorded indirectly through sediment temperature and moisture. Finally, measured CO2 concentrations were corrected for water vapor (lines 201-202).

• L167: I am not familiar with the mentioned sensor. What is its precision, compared to a Licor, Los Gatos, or Picarro gas analyzer, for example? At what frequency does this sensor measure during the five-minute closing time?

Reply: We used Sensirion SCD30 sensors, which can be programmed to measure at different frequencies; in our study, measurements were taken every 2–4 seconds during the five-minute chamber, or one hour chamber closure. While these sensors have a lower absolute accuracy (manufacturer-stated precision  $\pm$  30 ppm) compared to instruments such as Licor, Los Gatos, or Picarro analyzers ( $<\pm1$ ), the relative changes in CO2 concentrations are reliable for flux calculations, as only the change over time is relevant. The measurement principle follows Bastviken et al. (2015), who demonstrated that mini loggers can provide cost-efficient and reliable CO2 flux estimates in terrestrial and aquatic environments (see <a href="https://doi.org/10.5194/bg-12-3849-2015">https://doi.org/10.5194/bg-12-3849-2015</a> for accuracy

and reliability details). we have explicitly referenced their protocol in the revised manuscript (lines 212-213).

"The measurement approach used in this study follows Bastviken et al. (2015), who provide detailed information on logger preparation, sensor evaluation, calibration, and data processing in their manuscript and supplement."

• L174f: Explain why one measurement had a different sampling technique and measurement time. Was it a one-hour measurement time or a one-hour closing time?

Reply: For the present study, the 5-minute chamber measurements were used as a standard duration for addressing the CO2 research questions .The one-hour measurement method was also included, as it is compatible with our study and uses data from the internal CO2 sensor. This method involved a 1-hour chamber closure with 10-mL samples collected at 10 minutes intervals (0 min, representing ambient air, then 10, 20, 30, 40, 50 and 60 min). For the internal CO2 sensor, the only difference was the longer measurement period, making these data comparable to the 5-minute measurements. These one-hour measurements were primarily used to calibrate the CH4 sensor with field data, supplemented by an additional laboratory calibration. Manual gas samples analyzed by chromatography were used to measure CH4, CO2, and N2O. While these measurements are intended for future studies, the CO2 data from the logger were included in the present analysis as additional replicates, since fluxes were comparable between the 5-minute and 1-hour durations. Also, the CO2 from gas chromatography could be used to the reliability of sensor-based measurements. However, as our study focuses on fluxes, the absolute concentration is less critical than the relative changes over time, which are reliably detected by the SCD30 sensor for accurate flux calculations.

• L175: What was the size and the material of the vials or syringes? When during the one-hour closing time did you measure, and how much volume did you take? Were your chambers equipped with an overpressure valve?

Reply: We used 60 mL BD Plastipak syringes to withdraw gas from the chambers and transferred the samples into 5.9 mL Exetainer® vials (Labco). At each sampling time we collected 10 mL of chamber headspace. For the one-hour closure, six samples were taken at 10-min intervals (10, 20, 30, 40, 50 and 60 min). In addition, one extra sample of ambient air was taken at 0 min as a reference. The chambers were not equipped with an overpressure valve.

• L175f: What did you measure with the gas chromatograph? This is unclear to me, as it was never mentioned again in the manuscript.

Reply: The gas chromatograph was used to measure CH4, CO2, and N2O, as mentioned above. These measurements were not discussed further in the manuscript because they were not directly relevant to the results presented here.

We have clarified this here and revised the manuscript to provide a clearer explanation of the method while removing unnecessary references to these measurements.

• L178: Why did the CO2 sensor did not require a calibration?

Reply: The CO2 sensor (SCD30, Sensirion) measures concentrations in ppm and comes pre-calibrated by the manufacturer. Therefore, no additional calibration was required prior to deployment. A reference to the product datasheet (SCD30 CO2 and RHT Sensor Datasheet, Sensirion AG, 2020) has been added to the manuscript to clarify this point (line 200).

• L178f: What data did you use for the 3-point average, and why was there background noise?

Reply: The 3-point rolling average was applied to the raw CO2 concentration data to reduce background noise prior to flux calculation. Such smoothing is a common procedure to smooth the data before calculating fluxes, due to background noise arises of small, rapid fluctuations in the chamber headspace (pressure, temperature, RH changes) and sensor-internal measurement noise.

• L180: Why did you only use the last two to three minutes of each five-minute measurement period? What is the reference for Equation 1?

Reply: The initial 1-2 minutes of each measurement period were excluded due to increased signal noise likely caused by humidity and temperature fluctuations immediately after chamber closure. Flux values were derived from the data corresponding to the last 2–3 minutes of each 5-minute chamber closure to ensure more accurate linear flux estimates, as recommended for non-steady-state chamber measurements (Johannesson et al., 2024). The CO2 flux was then calculated based on the ideal gas law using Eq. (1) (Podgrajsek et al., 2014). Our equation is slightly modified, as we used the molar mass and report fluxes in mg C m-2 day-1.

• L182: Why did you choose to use the carbon unit for the fluxes? Additionally, you used hourly units here, but daily units throughout the manuscript.

Reply: While gas fluxes can be expressed in various units, we report them as CO2-C. In our study, fluxes are expressed in carbon units (mg C m-2 d-1), which allows direct comparison between CO2 and CH4 fluxes, following common practice in biogeochemical studies. We have also corrected this error; the fluxes are consistently reported in daily units throughout the manuscript (line 207).

• L190: Milli-Q is a brand name, not a water type.

Reply: You are right, we now use ultrapure water in the MS (line 217).

• L191: Why did you use 48 hours?

Reply: We followed the standard protocol, and sediment samples were dried at 105 °C for 48 hours to ensure complete water removal. The extended drying duration is particularly important for highly wet or fine-grained sediments, as shorter times may lead to underestimation of moisture content. This procedure aligns with ASTM D2216-19 (Doi:10.1520/D2216-19) and ensures accurate and reproducible results.

• L195: Please cite the references that used this as a proxy.

Reply: We have included the citation in the revised manuscript (line 223).

• L220: Table S2 is mentioned before Table S1. Please reconsider the order of the tables.

Reply: We have corrected the errors in the numbering and order of the tables.

• L221f: This sentence could be moved to the next subchapter, "Statistical (or data) analysis".

Reply: We appreciate the reviewer's suggestion. However, since this sentence refers to the data extraction procedure rather than the statistical analysis itself, we consider that it fits more naturally within the methodological section of dissolved organic matter characterization, rather than in the "Statistical analysis" subsection.

L224ff: Have you had tested your data for normality?

Reply: This has been modified in the revised manuscript: our data did not follow a normal distribution in all cases, as assessed by the Shapiro–Wilk test. Accordingly, we applied a Mann–Whitney test. Despite this adjustment, our results remain unchanged, and the explored groups continue to show the same significant differences, now confirmed with the appropriate statistical approach (lines 252-257)

• L259: The R version is important for repeatability.

Reply: We have added the R version used in the analysis to ensure repeatability (line 293).

• Figure 2: The unit in the y-axis label is missing a bracket. Why does the boxplots have different widths? Does the x-axis label mean Pond ID? To better illustrate the differences and support your results, I would reconsider the representation and categorize by other environmental variables.

Reply: All issues regarding Figure 2 have been corrected in the revised manuscript. Regarding the suggestion to categorize by other environmental variables, as the main purpose of this plot is to illustrate individual variability among ponds, we believe it would be less clear if the data were further aggregated. We have also added a bracket to indicate which Pond IDs correspond to each climate category.

• L263: Instead of "overall," I would say "on average" that all your ponds were a CO2 source, as you have also measured a few negative CO2.

Reply: The change has been implemented in the manuscript as recommended (line 294).

• L270: Both are red lines. Please be more precise in the description. What is the mean value of?

Reply: The change has been implemented in the manuscript to better clarify the graph (line 301).

• Table 2: I don't think the abbreviation T-FCO2 is necessary or correct here. I would rather use the term "CO2 fluxes" over the four columns on the right, including the unit. Where T-FCO2 and the unit are now, I would put the mean ± SD. I also recommend including the number of observations.

Reply: Thank you for your suggestion; the changes have been implemented in Table 2. However, the table has been moved to the Supplementary Material and renamed Table S1 by suggestion of the reviewer 2.

• Figure 3: In the graph, you used n.s., but in the caption, you explained the abbreviation NS. In the supplementary figures, you wrote that an absence indicates no significance. Be consistent.

Reply: The changes have been implemented throughout the manuscript and supplementary material, and we now consistently use "n.s." to indicate non-significant results in the figures.

• L299: Here, you wrote ".01"; in L302, you wrote "0.01" for p. Be consistent throughout the entire manuscript.

Reply: The p-value notation has been corrected for consistency throughout the manuscript.

• Figure 4: What are the R2 values of the linear regression lines?

Reply: The R2 values of the linear regression lines have been added to Figure 4 in the revised manuscript.

• L305: The caption of Figure 4 mentions a dashed line, but I cannot see one in the figure.

Reply: This element was removed in the latest version of the figure. We have now updated the caption accordingly to ensure consistency with the figure.

• Table 3: I miss the p values you mentioned in L302f.

Reply: The p-values mentioned in L302f have now been added to Table 1 in the revised manuscript.

• Figure 5: I'm not sure what to say about these trend lines. I don't trust them because they look like a point cloud with temperature differences and the influence of the edge effect.

Reply: To confirm that the interaction pattern is not an artifact of the model, we compared the fitted relationship with the raw data distribution. A supplementary figure (Fig Sr1) attached shows that the slopes of CO2 fluxes vs. sediment water content differ across temperature levels in the raw data, consistent with the interaction captured by our GLMM.

Fig Sr1. CO2 fluxes trends based on sediment temperature, with dots coloured and sized according to sediment water content. The figure was elaborated using raw data. Lower values are smaller and yellow, while higher values are larger and purple.

• L345: Sometimes there is a space between the number and "Celsius," and sometimes there isn't. Be consistent throughout the manuscript.

Reply: The errors mentioned have been corrected for consistency throughout the manuscript.

• Table 5: In the text, you always used SD, but in the tables, you used sd. Please be consistent and explain every abbreviation in the table caption. Use the same rounding for all emission values shown here. Since seasonality affects CO2 fluxes in your studied ponds and in most ecosystems outside the tropics, it would be helpful to know what season the reference values were measured.

Reply: All the mentioned changes have been implemented in the manuscript.

• L485: The data cannot be reviewed because they are unavailable. I recommend that the authors make the data easily accessible to everyone, not just upon request, for reasons of repeatability and reusability.

Reply: We thank the reviewer for this comment. The data are currently under embargo because they are part of ongoing doctoral theses. However, they will be made freely accessible once the embargo period ends, ensuring full repeatability and reusability at that time.

• The information about the R packages used is missing and can be added to the references or the methods chapter.

Reply: The R packages used have now been included in the Methods chapter, and their corresponding references have also been provided in the bibliography.

• Why did you use two appendices (A and B)?

Reply: We followed the egusphere journal's guidelines, which recommend the use of separate appendices (A and B) to organize supplementary material more clearly. However, if this format does not fully align with the journal's preferences or if another structure is recommended, we will be glad to adjust it accordingly.

• Table A2: Could you add more lines to better separate the categories in the left column and put each variable name in its own row? This would make the table easier to read.

Reply: We have reformatted Table A2 by adding additional lines to better separate the categories in the left column.

• Figure S1: Variable names, including units, are sometimes split into two lines. This makes the table difficult to read. Please reconsider this.

Reply: We believe the reviewer is referring to Table S1, after modification rename as Table S2. In response, we have reformatted the variable names, including their units, so that they now appear in a single line.

• Figure S2: In the manuscript, you called it "Pond ID"; here, on the x-axis, it is "Pond Code." Be consistent. The y-axis is missing a unit.

Reply: We thank the reviewer for pointing this out. The labels have been corrected for consistency, and the y-axis unit has now been included in Figure S2.

• Figure S3: Please include the number of observations for each box plot.

Reply: The number of observations for each box plot has now been included in Figure S3.

• Figure S4: There is a typo in the x-axis label: months.

Reply: The typo in the x-axis label of Figure S4 has been corrected.

• Figure S5. The sentence about the absence of an asterisk can be removed from the caption.

Reply: The sentence regarding the absence of an asterisk has been removed from the caption of Figure S5.

---

## Author Comment (AC2)

Thank you for taking the time to review our manuscript. We have carefully considered all of your comments and suggestions and revised the paper accordingly. We sincerely appreciate your thoughtful and constructive feedback, which has greatly helped us improve the clarity and overall quality of the manuscript. Below, we provide a detailed, point-by-point response to each of your comments, with your remarks shown in black and our replies in blue.

**Referee: 2**

**Comments to the Author(s)**

The manuscript presents two-season field measurements of CO2 emissions from air-exposed sediments in 30 ponds across Mediterranean and temperate climate zones. Based on the significant relationship between hydroperiod lengths and CO2 emissions in Mediterranean ponds in summer, the authors suggest that longer hydroperiods play a critical role in creating temporary conditions for higher CO2 emissions. Using various statistical approaches, they further identified key drivers of sediment CO2 emissions, including temperature and sediment contents of water and carbonate. The key findings from the well-designed study are novel and invite further study to elucidate the large temporal variability in CO2 emissions from ponds, which have been understudied compared to other freshwater systems. Despite the novelty and significance of the key findings, the manuscript shows weakness in linking and interpreting these findings, as well as a lack of detail in several areas, as described below. I hope my comments will help the authors improve the logical flow and clarity of the manuscript.

Reply: Thank you for your thorough suggestions and comments. We have revised the manuscript based on your feedback; please see our responses to your questions below.

**General comments**

**1. Hypotheses and data interpretation**

Although hydroperiods and sediment water contents are suggested as the primary controls on sediment CO2 emissions, descriptions across Introduction, Results, and Discussion appear not consistent, and in some cases contradictory.

First, hypotheses (2) and (3) need to provide more interrelated and mechanistic predictions. Higher sediment contents might be influenced more directly by more recent precipitation events (like 1-month or 1-week antecedent precipitation) than the yearly hydroperiod as considered here. Please provide a more detailed explanation of the relationship between hydroperiods and water contents. Any rationale for using hydroperiod rather than other drought indices would also be helpful.

Second, the findings shown in Fig. 4 indicate the significant relationship between hydroperiods and CO2 emissions only for Mediterranean ponds in summer, and the significance appears controlled by a few sites with very long hydroperiods. However, this hydroperiod effect is emphasized too much across the R & D sections, with some of them having inconsistent connotations: for instance, refer to L 275-276 ("Mediterranean ponds exhibited higher air and sediment temperatures, shorter hydroperiods, typically drying in summer. They also showed lower sediment water content, and reduced macrophyte coverage, consistent with an earlier drying period."). Please check the consistency of descriptions across R & D (sections 4.1 and 4.2 appear to address two separate stories regarding the hydroperiod

effect) to provide a more coherent explanation for the relationship between hydroperiods and sediment water contents.

Reply: Thanks you for the comments. Here we provide a detailed response to all the comments in order:

• First, we selected hydroperiod length as a key explanatory variable because it provides a simple, readily measurable proxy for the cumulative effects of the preceding wet phase. Unlike single-point measurements, hydroperiod length integrates a suite of biotic and abiotic processes that occur during inundation (accumulation and transformation of organic carbon, macrophyte growth and senescence, and changes in nutrient loading), which can influence sediment properties and subsequent CO2 dynamics during the dry phase. Therefore, using hydroperiod length allows us to capture these influences on carbon processing that occur before measurements, without requiring extensive, often unavailable, time-series data, such as precipitation data or water table levels. Short-term data as you suggest (e.g., 1-week to 1-month) are more directly reflected by site-specific sediment measurements, such as sediment water content. Although we found a moderate correlation between hydroperiod length and sediment water content (r = .47; Fig. B1), hydroperiod length captures broader conditions that are not completely explained by sediment water content (Fig. Sr2) and the combination of sediment water content and temperature (Fig. Sr3).

Mediterranean
Temperate

Season
Summer Autumn

Hydroperiod length (months)

Fig Sr2. Relationship between hydroperiod length (months) and sediment water content (%) in ponds from Mediterranean and Temperate regions, separated by season (Blue= Summer and orange=Autumn). Each point represents the mean per pond and season, and the lines show the linear trend with its confidence interval (shaded area).

Fig. Sr3 Relationship between CO2 fluxes and hydroperiod. To examine the drivers of emissions, dots represent the mean CO2 flux per pond and season. Dot size corresponds to sediment water content, and colour indicates sediment temperature, ranging from blue (low) to red (high). Solid lines represent trends for summer, and dashed lines represent trends for autumn.

We have expanded the explanation in the Introduction (lines 94–96) and added a methodological justification in the Methods section (lines 145-148) to clarify this approach for readers.

• Second, we have revised the manuscript to improve clarity and coherence in the description and interpretation of results related to hydroperiod (lines 414-417). We have included modifications along results and discussion to improve the new revised manuscript (e.g. line 457).

**2. Realigning paragraphs**

Although the manuscript was easy to follow on a sentence-by-sentence level, the use of very long or several scattered short paragraphs made it difficult to grasp the overall logical structure. In the Introduction, for example, the page-long initial paragraph is followed by five short paragraphs. A thorough revision of the manuscript is recommended to reorganize the long and short paragraphs in accordance with a coherent logical flow.

Reply: We appreciate the reviewer's observation regarding paragraph structure and logical flow. The Introduction and other sections have been thoroughly revised to improve readability and coherence.

**3. Clarity of tables and figures**

There are numerous missing or inaccurate details that could be improved through careful revision. Please refer to the specific comments below.

Reply: We have carefully revised the tables and figures in accordance with the suggestions and notifications received.

**Specific comments**

• Title: A slight change would enhance clarity: for example, Drivers of CO2 emissions during the dry phase "in" Mediterranean and Temperate ponds or Drivers of CO2 emissions "from" Mediterranean and Temperate ponds "during the dry phase".

Reply: We have modified the title following the first suggestion, and it now reads: "Drivers of CO2 emissions during the dry phase **in** Mediterranean and Temperate ponds."

• Line (L) 17: sources of carbon (or CO2)?

Reply: While "sources of carbon" is more general and correct, we agree that given the focus of our study it is more appropriate to specify "CO2" here. The sentence has been revised accordingly (Line 17).

• L 17 "remain largely overlooked": This statement overlooks the decadal research on this topic.

Reply: We agree that stating "remain largely overlooked" may be too categorical. The sentence has been revised to specify the ecosystem targeted in our study, now reading: "Emissions of CO2 during their dry phases remain relatively understudied in some inland waters, such as pond" (Line 17).

• L 25: "the" interaction

Reply: The error has been corrected (Line 26).

• L 35-38: Please provide some estimates of CO2 and CH4 emissions from ponds to describe their role more quantitatively.

Reply: We have now added representative CO2 emission values from the literature as a reference in lines 40-43. CH4 emission values were not included, as the focus of the present study is on CO2 emissions.

• L 85: Please define "hydroperiod length".

Reply: In this study, hydroperiod refers to the duration of time each pond retained water prior to the dry phase, quantified as the number of months with water surface during the 12-month period preceding the last autumn sampling (conducted between late September and November). We have now included a definition of hydroperiod length in the manuscript for better clarification in lines 94-95.

"Hydroperiod length (i.e., the duration of water presence prior the dry phase in a pond throughout the year)"

• L 96: Without the above-mentioned definition, it is difficult to understand "shorter hydroperiods leading to lower emissions due to reduced sediment water content".

Reply: We have now explicitly defined hydroperiod length prior to this section (L96) in lines 94-95. We also rephrased the sentence to clarify this point in lines 107-108.

• L 99: Can you illustrate "conservation status" using an example?

Reply: We thank the reviewer for the suggestion. An example illustrating "conservation status" has been added to the manuscript (lines 111-113), describing some characteristics of well-conserved ponds.

"better conservation status (e.g., clear water with turbidity < 5 NTU, extensive native emergent vegetation, and  $\ge 50$  % hydrophytic plant cover, particularly vascular submerged species or charophytes covering > 75 % of the pond bottom), will exhibit greater CO2 emissions due to increased vegetation senescence during the dry phase."

• L 117: Did 23 sites also include semi-permanent and permanent ponds? In the latter case, the described bare sediment would be contradictory to the definition of permanent ponds (L 110).

Reply: Yes, they did. We would like to clarify that the classification of ponds as temporary, semi-permanent, or permanent is based on a three-year record. However, as explained in lines 124-128, during the sampling year (2022), extreme drought conditions caused even some semi-permanent and permanent ponds to dry almost completely. This resulted in large areas of exposed sediment, allowing us to measure CO2 emissions under conditions that can occur in more permanent ponds during extreme dry years. We added a sentence in the manuscript to clarify this aspect (lines 127-128).

- L 130 "water presence": Do you mean rainy days or literal water presence in ponds?

  Reply: We refer to the literal presence of water in the ponds, not rainfall. We clarified this point in the manuscript (lines 145-146).
- L 154 (throughout the manuscript): not Chlorophyll a, but chlorophyll a
   Reply: Thank you for noting this. The error has been corrected throughout the revised manuscript.
- L 167: Please provide key details on the chamber design, including the used material, size, ventilation, etc.

Reply: We used static, opaque chambers with a surface area of 0.075 m2 and a total volume of 8 L (diameter 345 mm, height 160 mm). The chambers were made of polypropylene (PP) plastic, and no additional materials such as tubes or pumps were used during the measurements. We have added the requested details on the chamber design, including material, size, and ventilation in the manuscript (lines 185-192).

• L 174-177: It would provide useful information for assessing the accuracy of sensor data if you compare sensor and additional GC measurements.

Reply: We thank the reviewer for the suggestion.  $CO_2$  samples measured by GC are available for comparison with the sensors; however, as our study focuses on fluxes, the absolute concentration is less critical than the relative changes over time. The manufacturer-stated precision ( $\pm$  30 ppm) ensures that relative changes in  $CO_2$  concentrations are reliable for flux calculations, we followed the methodology of

Bastviken et al. (2015), who demonstrated that mini loggers provide cost-efficient and accurate CO2 flux estimates in terrestrial and aquatic environments (see <a href="https://doi.org/10.5194/bg-12-3849-2015">https://doi.org/10.5194/bg-12-3849-2015</a> for accuracy and reliability details). We have explicitly referred to this study in the manuscript to clarify and support the validity of the methodological approach followed (lines 212-213).

• L 195: Please provide a relevant reference for this carbonation estimation.

Reply: We have included the reference in the revised manuscript (line 223).

• L 224: How did you test the normal distribution of your datasets?

Reply: We tested the normality of our datasets using the Shapiro–Wilk test. However, some data did not meet the assumption of normality. Accordingly, we reanalyzed the data using a non-parametric approach (Mann–Whitney test). Despite this adjustment, our results remained unchanged, showing the same significant differences, now confirmed with the appropriate statistical approach. This information has been added in lines 252-257.

• L 265: Are these negative values from partially water-flooded sediments where phytoplankton take up CO2? Please elaborate on the site characteristics and discuss the meaning of these values (if outside measurement error ranges).

Reply: We observed negative CO2 fluxes only in a few cases, representing approximately 4% of the total fluxes reported (10 out of 249). Typically, there was only one negative measurement per pond, except for one pond (SP044), which showed two. The mean  $\pm$  SD of these negative fluxes was -257.6  $\pm$  191.3 mg C m-2 d-1 (N = 10; min = -611.2, max = -1.4; median = -244.7), with no significant difference between seasons (Summer: N = 5,  $-275.5 \pm 259.1$  mg C m-2 d-1; Autumn: N = 5, - $239.7 \pm 120.2 \text{ mg C m}^{-2} \text{ d}^{-1}$ ; t = -0.28, df = 8, p = 0.787). These sites did not show any consistent relationship with the main drivers of CO2 emissions, such as sediment temperature or water content. The magnitude of these negative fluxes is consistent with values reported in other studies using closed chambers. For instance, Keller et al. (2019) reported -324 mg C m-2 d-1, while Ma et al. (2013) observed -290 and -436 mg C m-2 d-1 in under-canopy and inter-plant spaces, respectively. Similarly, Koschorreck et al. (2022) found fluxes ranging from -1,440 to 13,620 mg C m-2 d-1, with negative values representing 6% of all measurements. Since all measurements in our study were conducted using opaque chambers, and we measured fluxes in bare sediments, it is unlikely that these negative values are due to CO2 uptake by residual phytoplankton, plants, or cryptobiotic crusts. Therefore, these negative fluxes most likely reflect physico-chemical processes in the sediments, probably linked to inorganic reactions (Ma et al., 2013; Marcé et al., 2019). This explanation has been incorporated into the Discussion section of the revised manuscript (lines 468-474).

• L 284-286: Please clarify whether you are talking about the proportion of each component based on unit mass of sediment or DOC.

Reply: We appreciate the reviewer's observation. The comparison refers to the relative proportion of each PARAFAC component within the total fluorescent DOC signal,

rather than to values normalized by sediment mass. We have clarified this in the revised manuscript (lines 309-310).

• L 288-293: These sentences are good examples of unnecessary separation mentioned before.

Reply: The unnecessary separation between sentences has been removed in the revised version.

• L 300: Given the significance of the hydroperiod effect, it would be helpful to elaborate more as to how "the effect of hydroperiod was season-specific and climate-dependent" as displayed in Fig 4.

Reply: We have clarified the description of the hydroperiod effect in lines 331-337 to better explain how it was season-specific and climate-dependent. Additionally, the corresponding p-values have been added to Table 1 to provide more detailed information.

• L 301: Was the summer trend also significant for the temperate sites?

Reply: The summer trend was not significant for the Temperate sites. To clarify this point, we have added the corresponding p-values in the revised manuscript in Table 1 (lines 340). When both climatic regions are considered together, the overall trend is significant; however, when analyzed separately, the significant effect is only observed for the Mediterranean ponds.

• L 320-330: In a sense, this part seems secondary but covers the bulk of section 3.2. More space could be saved for more relevant drivers.

Reply: This section (lines 351-355) has been considerably reduced in the revised manuscript in accordance with your comment, to focus more on the most relevant drivers.

• L 354 "all ponds emitted CO2 during the dry phase": This statement is contradictory to the result descriptions (Fig. 2).

Reply: Only a few measurements (10 out of 249) showed negative CO2 fluxes, but these were minor and isolated occurrences, with overall flux patterns indicating CO2 emission across ponds. To more accurate statement, we modify the manuscript in line 383.

• L 357: "shaped" or "was shaped by"?

Reply: We thank the reviewer for this comment. We have retained the active form "shaped" in the manuscript, as it accurately reflects the causal relationship described (line 386).

• L 445: It would help readers to compare the magnitudes of plant uptake vs. CO2 emissions if you provide some literature values estimating plant C uptake.

Reply: We thank the reviewer for this suggestion. While we agree that it would be valuable to provide literature values for plant C uptake to compare with CO2

emissions, such values strongly depend on the macrophyte species and the characteristics of the specific waterbody. Due to this high variability, it was not possible to provide reliable estimates of species-specific CO2 uptake for our study system. However, to give readers a quantitative perspective, we have now included literature-reported ranges of carbon burial in small ponds depending on vegetation cover (Taylor et al., 2019), as well as a comparison of CO2 fluxes measured in bare and vegetated areas of wetlands under both light and dark conditions to assess the potential of aquatic vegetation to offset CO2 emissions (Sharma et al., under revision), and information on net ecosystem exchange (NEE) from Madaschi et al. (2025). These additions are included in lines 485-492.

L 417: Fig 5 shows the generally highest levels of CO2 emissions across the highest temperature ranges.

Reply: We agree with the reviewer. The corresponding correction has been included in the manuscript in line 451.

• L 460 "ponds with more permanent hydroperiod": This is quite confusing, given your descriptions of your sites. Did you mean simply "longer hydroperiod"?

Reply: We have modified the sentence, now reading "longer hydroperiods" (line 510).

• Fig 1 caption: Countries "are"

Reply: The figure caption has been corrected in the revised manuscript (line 120).

• Fig 2: Please complete the vertical axis title with the second parenthesis.

Reply: The error has been corrected in the revised version of the manuscript.

• Table 2: If this displays the same data as Fig 2, please think about removing or revising it to avoid double presentation.

Reply: Table 2 has been moved to the Supplementary Material (Table S1) for readers interested in specific details on CO2 fluxes.

• Fig 4: Please indicate the significance levels for the depicted regressions. It would be easier to find out the significance if only significant regressions were shown as regression lines.

Reply: The significance levels of the regressions are now included in Table 1 to indicate which relationships are statistically significant, complementing Fig. 4.

• Tables 3, 4, 5: Please explain in the caption the abbreviations including SE, df, CL, AIC, BIC, and CI.

Reply: Explanations of all abbreviations (SE, df, CL, AIC, BIC, and CI) have been added to the captions of all tables.

• Figure 5: What is ORQ? Are all the depicted trends statistically significant?

Reply: ORQ (Ordered Quantile normalization) is a data transformation applied to meet the assumptions of normality, This has been added to the Figure 5 caption. The partial trends shown in Figure 5 were evaluated using 95% confidence intervals of model-predicted values (via the R package visreg). While the confidence intervals for the three sediment temperature levels (9.4 °C, 18.2 °C, and 27.7 °C), cross zero, significance is assessed at the model level. In this GLMM, the interaction between sediment water content and sediment temperature is statistically significant based on the fixed-effects confidence intervals.

• Table 5: What about showing the employed models in a separate column?

Reply: We have clarified the model used in the table caption for better readability. Since all estimates come from the same model, adding a separate column in Table 3 was deemed necessary for clarity or aesthetics (line 378).

• Table "6" (page 18): Please also correct the unnecessary values below the decimal point.

Reply: Thank you for noting these issues. The table number has been corrected, and the unnecessary decimal values have been removed in the revised version of the manuscript.

---

## Author Response (AR1)

Thank you for taking the time to consider our manuscript. We have addressed all the comments in the revised manuscript and are grateful for the careful and constructive feedback. The suggestions have substantially improved the clarity and quality of the paper. Below we provide a point-by-point response to each comment, with your comments shown in black and our responses in blue.

**Referee: 1**

Comments to the Author(s)

The study examines $CO_2$ fluxes of dry sediments from 14 temperate and 16 Mediterranean ponds in Europe during their dry phase in summer and/or autumn of 2022. These fluxes were measured using chambers. Additionally, sediment and water analyses were conducted to characterize the sites and to investigate the differences.

I read the manuscript with great interest and believe that the study is promising, but that some improvements are necessary.

Reply: Thank you for your time and effort in providing us with constructive feedback to improve our manuscript.

General comments or questions:

- In the abstract, the methodology for measuring $CO_2$ fluxes is missing.

  Reply: We have now included a brief description of the methodology used for measuring $CO_2$ fluxes in the abstract in lines 21-22. We aimed to keep it concise, as abstracts typically require a short overview.

  "We measured $CO_2$ emissions from air-exposed sediments using closed static chambers equipped with internal mini-loggers in 30 ponds across Mediterranean and Temperate regions."

- How exactly and to what extent does your study fill the knowledge gap you mentioned in the introduction?

  Reply: We acknowledge the reviewer's comment. Our study contributes new data on $CO_2$ emissions from ponds during the dry phase, that in comparison with other ecosystems is less well represented. Our study highlights the importance of incorporating seasonal frameworks that account for the main local factors controlling $CO_2$ fluxes, such as temperature and water content. Moreover, it introduces hydroperiod length as a first step toward a more integrated understanding of both wet and dry phases, elucidating how transitions from permanent to temporary ponds can affect $CO_2$ dynamics. However, we recognize that this study represents only an initial step, and further research is needed to fully unravel the complexity and variability of these dynamic ecosystems.

- In the introduction, a concluding sentence on how the questions will be answered is missing.

Reply: We have now added a concluding sentence in the introduction to clarify how the research questions are addressed in lines 93-96. We believe that this addition improves the flow and clarity of the section.

"For this reason, our study aims to address this gap by identifying the main drivers of $CO_2$ fluxes during dry periods and examining how the preceding wet phase, in terms of hydroperiod length (i.e., the duration of water presence prior the dry phase in a pond throughout the year) influence them, through a comparison of ponds from contrasting climatic regions across two seasons."

- The coordinates of the sampling or measurement points within the sites, the sampling dates, and the names of the studied ponds are missing.

  Reply: We have added a new table in the Appendix (Table A1) that provides the coordinates, the names of the studied ponds, and the corresponding sampling dates (line 531).

- The chamber description still lacks some information (see the detailed comments).

  Reply: We added the requested information in the new version of the manuscript (Section 2.3).

- For me, it is unclear how often or at what frequency the $CO_2$ measurements were conducted in each season. Were there temporal replicates?

  Reply: We have clarified the frequency of $CO_2$ measurements for each season. Specifically, measurements were conducted once per season in each pond, and no temporal replicates were performed. This clarification has been included to provide a more accurate description of the sampling design (lines 182-183).

  "Measurements were conducted once per season at each pond during daytime (08:00–19:00 h), with no temporal replicates within the same season (Table A1; S1)."

- In the manuscript, there is no information on whether the data requirements for the statistical tests used (e.g., ANOVA or t-test) are met.

  Reply: We thank the reviewer for this suggestion. As $CO_2$ emissions did not follow a normal distribution in some cases, we revised the analyses using non-parametric Mann–Whitney tests (Wilcoxon rank-sum test; wilcox_test function, rstatix package; (assambara, 2023). Normality was assessed prior to analysis using Shapiro–Wilk tests (shapiro.test function, stats package). Importantly, the main patterns and differences between groups remain unchanged, while the revised approach provides a more robust statistical framework. We added the information in lines 255-258.

"We used the non-parametric Mann–Whitney test (Wilcoxon rank-sum test) to compare overall $CO_2$ emissions across climates (Mediterranean, Temperate) and seasons (Summer, Autumn) (Hypothesis 1), based on the average of multiple measurements in each pond (wilcox_test function in rstatix package (Kassambara, 2023)). Before analysis, data were assessed for normality using Shapiro–Wilk tests (shapiro.test function in stats package (R Core Team, 2024))."

- I miss a conclusion chapter.

  Reply: We thank the reviewer for this comment. A conclusion chapter has now been added to the manuscript (lines 523-529).

  "Our results highlight the need to integrate $CO_2$ emissions across all stages of the dry season to achieve accurate estimates of fluxes in ponds. Although no significant differences in emissions were observed among climatic regions, key drivers such as hydroperiod length, sediment temperature, and sediment water content are inherently linked to climate. Moreover, ponds with better conservation status emitted more $CO_2$ during the dry phase; however, a comprehensive integration with emissions from the wet phase is still required. Understanding these dynamics is crucial for predicting carbon fluxes in pond ecosystems under future climate and land-use scenarios."

- The graphics are slightly pixelated, and the image quality could be improved.

  Reply: We will ensure the images meet the correct pixel standards recommended on the journal's webpage.

  Detailed comments and suggestions:

- Abstract: The methodology is missing

  Reply: As mentioned above, a brief description of the methodology has now been included in the abstract. We kept it concise to maintain the abstract's brevity, while providing enough information to understand the general approach (lines 21-22).

- L23ff: What do you mean by hydroperiod?

  Reply: In this study, hydroperiod refers to the duration of time each pond retained water prior to the dry phase, quantified as the number of months with water surface during the 12-month period preceding the last autumn sampling (conducted between late September and November). We have added a brief description in the introduction to clarify this term (lines 94-95).

  "Hydroperiod length (i.e., the duration of water presence prior the dry phase in a pond throughout the year)"

- L26: c.?

  Reply: The abbreviation "c." stands for "circa", meaning "approximately". To improve clarity, we have replaced it with approximately 27 ºC in the manuscript (line 27).

- L37f: It would be good to specify the $CO_2$ emission value of ponds.

  Reply: We have now added some representative $CO_2$ emission values from the literature as a reference in lines 40-43.

  "However, reported $CO_2$ emissions from ponds are highly variable, ranging from hundreds to several thousand mg C $m^{-2}$ $d^{-1}$. For instance, ponds of similar size (< 0.001 and 0.001–0.01 $km^2$) reported by Holgerson and Raymond, (2016) emitted on average 254 and 422 mg C $m^{-2}$ $d^{-1}$, respectively, whereas exposed pond sediments reported by Keller et al. (2020) range from -73 to 11765 mg C $m^{-2}$ $d^{-1}$."

- L104f: Instead of spanning latitudes or longitudes, it would be better to add the actual coordinates of some sites and refer to the appendix.

  Reply:  The coordinates of the sites have now been included in Table A1 in the Appendix, as recommended (line 531).

- Figure 1: Since graphs a) and b) are already zoomed in, it would be helpful to include an overview graph with all sites, countries, climate regions, etc.

  Reply: We thank the reviewer for this suggestion. An overview graph including all sites and countries, providing a better zoomed-out view, has been added to the revised manuscript (line 118).

[Figure]

Figure 1: Geographic locations of the studied ponds. Left panel: map created using Stadia Maps outdoors basemap and OpenStreetMap data. Map data © OpenStreetMap contributors, © Stadia Maps (https://stadiamaps.com/). Ponds are highlighted in red. Right panel: map showing ponds categorized by hydroperiod: temporary (dark red), semi-temporary (yellow), and permanent (dark blue). Countries are colour-coded according to their climate regions: Mediterranean (orange) and Temperate (light blue).

- It would be interesting to categorize the sites in Figure 1 according to the dry or wet phase definition written in L109 ff.

Reply: The sites in Figure 1 have now been categorized according to the categorical definition described in L109 ff, as suggested. Indicated in the figure attached above.

- Method chapter: Please use the same description style for each instrument. Currently, it differs. You do not have to repeat it if the instrument is already mentioned.

Reply: The Methods section has been revised to ensure a consistent description style for all instruments, standardizing the information to include model and brand, and avoiding repetition when an instrument had already been mentioned.

- L117: What do you mean by "dry fluxes"? I also recommend using $N$ instead of n for the number of observations/samples throughout the manuscript.

Reply: In the literature, "dry fluxes" refers to $CO_2$ fluxes measured from exposed (non-flooded) sediments after the waterbody is no longer inundated. We have clarified this in the revised manuscript (Line 131).

- L123ff: Why were 40-year averages of annual temperatures and precipitation used instead of the more common 30-year averages for the climatic description of sites?

Reply: We used 40-year averages because the climatic database employed in our study (fifth generation of ECMWF atmospheric reanalysis of the global climate (ERA5) of Copernicus Climate Change Service (C3S) (Wouters, 2021)) provides reference values for the period 1978–2018, which corresponds to 40 years. We considered it appropriate to use the available period rather than artificially truncating the dataset, as the longer time frame reduces the influence of short-term anomalies while still representing contemporary climatic conditions. For year of deviance, we included the year of sampling 2022. So, we can capture the trend and the most accurate of the year.

- L130: You have finally defined what you mean by "hydroperiod length," although the term was already mentioned before. It would have been helpful to define it the first time it was mentioned.

Reply: As mentioned, brief description of "hydroperiod length" (i.e., the duration of water presence in a pond over the course of a year) has now been added prior to its definition in the Methods section, to clarify the term and improve readability (lines 94-95).

- L157: Why did you use a filter size of 0.7 μm to obtain the dissolved fraction instead of, for example, 0.45 μm?

Reply: We used 0.7 μm GF/F filters because this pore size efficiently separates the dissolved fraction while retaining phytoplankton and larger particles, and reduces clogging compared with finer 0.45 μm filters. Moreover, the use of

0.7 µm GF/F filters has been historically a common approach in oceanography and limnology, allowing comparability with previous work and consistency in long-term datasets (DFO, 2015; Wetzel and Likens, 2000).

DFO (2015): Fisheries and Oceans Canada, Reference Manual for Limnological Analyses, Publications.gc.ca collection, Fs94-167.

Wetzel, R. G. and Likens, G. E. (2000): Limnological Analyses, 3rd edn., Springer-Verlag, New York, 429 pp., https://doi.org/10.1007/978-1-4757-3250-4.

- L165ff: This subchapter is missing important information about the chamber measurement (e.g., whether it is a through-flow or static system), the chamber type (transparent or opaque), the chamber size (area, height, and volume), and the chamber material. In addition, did you use any additional materials during the measurement, such as tubes or a pump? What time of day did you conduct the chamber measurements at the four to eight spots per pond? How many measurement days were there per season at each pond, and how often were they conducted? Did you correct the measured $CO_2$ concentrations for water vapor?

  Reply: We have added the requested methodological details in the revised manuscript. In response to the questions:
  We used static, reflective chambers with a surface area of 0.075 $m^2$ and a total volume of 8 L (diameter 345 mm, height 160 mm). The chambers were made of polypropylene (PP) plastic and covered with aluminium tape, with small fan inside at the top to recirculate the air and prevent stratification, but there was no airflow through the chamber ( no additional tubes or pumps were used during the measurements) (lines 184-191).
  Measurements were conducted during daytime between 08:00 and 19:00. Since our study design did not include temporal replicates, only one measurement campaign was carried out per season in each pond (lines 181-183). Because the chambers are reflective, preventing light from entering, diurnal variations are recorded indirectly through sediment temperature and moisture. Finally, measured concentrations were corrected for water vapor (lines 202-203).

- L167: I am not familiar with the mentioned sensor. What is its precision, compared to a Licor, Los Gatos, or Picarro gas analyzer, for example? At what frequency does this sensor measure during the five-minute closing time?

  Reply: We used Sensirion SCD30 sensors, which can be programmed to measure at different frequencies; in our study, measurements were taken every 2–4 seconds during the five-minute chamber, or one hour chamber closure. While these sensors have a lower absolute accuracy (manufacturer-stated precision ± 30 ppm) compared with instruments such as Licor, Los Gatos, or Picarro analyzers (< ±1) . However, as we are calculating fluxes it is only the relative changes in $CO_2$ concentrations that are crucial for reliable for flux calculations, as only the change over time is relevant. The measurement

principle follows Bastviken et al. (2015), who demonstrated that mini loggers can provide cost-efficient and reliable $CO_2$ flux estimates in terrestrial and aquatic environments (see https://doi.org/10.5194/bg-12-3849-2015/ for accuracy and reliability details). we have explicitly referenced their protocol in the revised manuscript (lines 215-216). However, we also compared the sensor performance explained in the next point.

"The measurement approach used in this study follows Bastviken et al. (2015), who provide detailed information on logger preparation, sensor evaluation, calibration, and data processing in their manuscript and supplement."

- L174f: Explain why one measurement had a different sampling technique and measurement time. Was it a one-hour measurement time or a one-hour closing time?

  Reply: For the present study, the 5-minute chamber measurements were used as a standard duration for addressing the $CO_2$ research questions. The one-hour measurement method was also included, as it is compatible with our study and uses data from the internal $CO_2$ sensor. This method involved a 1-hour chamber closure with 10-mL samples collected at 10 minutes intervals (0 min, representing ambient air, then 10, 20, 30, 40, 50 and 60 min). For the internal $CO_2$ sensor, the only difference was the longer measurement period, making these data comparable to the 5-minute measurements. These one-hour measurements were primarily used to calibrate the $CH_4$ sensor with field data, supplemented by an additional laboratory calibration. Manual gas samples analyzed by chromatography were used to measure $CH_4$, $CO_2$, and $N_2O$. While these measurements are intended for future studies, the $CO_2$ data from the logger were included in the present analysis as additional replicates, since fluxes were comparable between the 5-minute and 1-hour durations. Also, the $CO_2$ from gas chromatography could be used to the reliability of sensor-based measurements. However, as our study focuses on fluxes, the absolute concentration is less critical than the relative changes over time, which are reliably detected by the SCD30 sensor for accurate flux calculations.

  Indeed, to evaluate the accuracy of the Sensirion $CO_2$ sensors used in the study, we provide a comparison between sensor readings with gas chromatograph (GC) measurements for two of the sensors (S1 and S2), which were used for all GHG measurements in Spanish ponds.

For raw absolute concentrations, Sensirion sensors underestimated absolute $CO_2$ concentrations by ~7–11% relative to the gas chromatograph the GC–sensor relationship was strongly linear ($R^2 = 0.96$–$0.98$) but with slopes slightly below 1 (S1 = 0.93; S2 = 0.89).

[Figure]

Because fluxes depend on the rate of concentration change ($\Delta CO_2/\Delta t$), we also compared GC and sensor data after subtracting the air baseline. This correction improved agreement for both sensors, with slopes of 0.97 (S1) and 0.93 (S2). This confirms that the proportional relationship between GC and sensor measurements is strong and that relative changes in $CO_2$ (the basis for flux calculations) are reliably captured.

[Figure]

- L175: What was the size and the material of the vials or syringes? When during the one-hour closing time did you measure, and how much volume did you take? Were your chambers equipped with an overpressure valve?

  Reply: We used 60 mL BD Plastipak syringes to withdraw gas from the chambers and transferred the samples into 5.9 mL Exetainer® vials (Labco). At each sampling time we collected 10 mL of chamber headspace. For the one-hour closure, six samples were taken at 10-min intervals (10, 20, 30, 40, 50 and 60

min). In addition, one extra sample of ambient air was taken at 0 min as a reference. The chambers were not equipped with an overpressure valve.

- L175f: What did you measure with the gas chromatograph? This is unclear to me, as it was never mentioned again in the manuscript.

  Reply: The gas chromatograph was used to measure $CH_4$, $CO_2$, and $N_2O$, as mentioned above. These measurements were not discussed further in the manuscript because they were not directly relevant to the results presented here. We have clarified this here and revised the manuscript to provide a clearer explanation of the method while removing unnecessary references to these measurements (lines 196-198).

- L178: Why did the $CO_2$ sensor not require a calibration?

  Reply: The $CO_2$ sensor (SCD30, Sensirion) measures concentrations in ppm and comes pre-calibrated by the manufacturer. The sensors were configured following manufacturer recommendations, including forced recalibration (we used 410 ppm), altitude compensation (60 m), atmospheric pressure compensation (1000 mbar), and a fixed measurement interval, ensuring stable operation across deployments. The calibration for one-point $CO_2$ calibration was performed using a custom Arduino script (Code attached for reviewers' revision). This information and the reference to the product datasheet (SCD30 $CO_2$ and RHT Sensor Datasheet, Sensirion AG, 2020) has been added to the manuscript to clarify this point (lines 199-202).

  Specific doubt related to this question is addressed at the end of the file in Note in the editor's decision

- L178f: What data did you use for the 3-point average, and why was there background noise?

  Reply: The 3-point rolling average was applied to the raw $CO_2$ concentration data to reduce background noise prior to flux calculation. Such smoothing is a common procedure to smooth the data before calculating fluxes, due to background noise arises from small, rapid fluctuations in the chamber headspace (pressure, temperature, RH changes) and sensor-internal measurement noise. This is a common approach with sensors of all types used in the wild. For example, some sensor systems e.g. the Exo sonde produced by YSI and used very widely has a running mean applied to the sensor readings before reported them to the user, so we never see the actual raw data from these types of sensors. Here we applied it, in part as the measurements were so frequent and it works better when fluxes are lower.

- L180: Why did you only use the last two to three minutes of each five-minute measurement period? What is the reference for Equation 1?

Reply: The initial 1-2 minutes of each measurement period were excluded due to increased signal noise likely caused by humidity and temperature fluctuations immediately after chamber closure. Flux values were derived from the data corresponding to the last 2–3 minutes of each 5-minute chamber closure to ensure more accurate linear flux estimates, as recommended for non-steady-state chamber measurements (Johannesson et al., 2024). The $CO_2$ flux was then calculated based on the ideal gas law using Eq. (1) (Podgrajsek et al., 2014). Our equation is slightly modified, as we used the molar mass and report fluxes in mg C $m^{-2}$ $day^{-1}$. We include the information and reference in the revised manuscript (lines 203-210).

- L182: Why did you choose to use the carbon unit for the fluxes? Additionally, you used hourly units here, but daily units throughout the manuscript.

  Reply: We have also corrected this error; the fluxes are consistently reported in daily units throughout the manuscript (line 211). While gas fluxes can be expressed in various units, we report them as $CO_2$-C. In our study, fluxes are expressed in carbon units (mg C $m^{-2}$ $d^{-1}$), which allows direct comparison between $CO_2$ and $CH_4$ fluxes, following common practice in biogeochemical studies.

- L190: Milli-Q is a brand name, not a water type.

  Reply: You are right, we now use ultrapure water in the MS (line 220).

- L191: Why did you use 48 hours?

  Reply: We followed the standard protocol, and sediment samples were dried at 105 °C for 48 hours to ensure complete water removal. The extended drying duration is particularly important for highly wet or fine-grained sediments, as shorter times may lead to underestimation of moisture content. This procedure aligns with ASTM D2216-19 (Doi:10.1520/D2216-19) and ensures accurate and reproducible results.

- L195: Please cite the references that used this as a proxy.

  Reply: We have included the citation in the revised manuscript (line 226).

- L220: Table S2 is mentioned before Table S1. Please reconsider the order of the tables.

  Reply: We have corrected the errors in the numbering and order of the tables.

- L221f: This sentence could be moved to the next subchapter, "Statistical (or data) analysis".

  Reply: We appreciate the reviewer's suggestion. However, since this sentence refers to the data extraction procedure rather than the statistical analysis itself, we consider that it fits more naturally within the methodological section of

dissolved organic matter characterization, rather than in the "Statistical analysis" subsection.

- L224ff: Have you had tested your data for normality?

Reply: This has been modified in the revised manuscript: our data did not follow a normal distribution in all cases, as assessed by the Shapiro–Wilk test. Accordingly, we applied a Mann–Whitney test. Despite this adjustment, our results remain unchanged, and the explored groups continue to show the same significant differences, now confirmed with the appropriate statistical approach (lines 255-258).

- L259: The R version is important for repeatability.

Reply: We have added the R version used in the analysis to ensure repeatability (line 293).

- Figure 2: The unit in the y-axis label is missing a bracket. Why does the boxplots have different widths? Does the x-axis label mean Pond ID? To better illustrate the differences and support your results, I would reconsider the representation and categorize by other environmental variables.

Reply: All issues regarding Figure 2 have been corrected in the revised manuscript. Regarding the suggestion to categorize by other environmental variables, as the main purpose of this plot is to illustrate individual variability among ponds, we believe it would be less clear if the data were further aggregated. We have also added a bracket to indicate which Pond IDs correspond to each climate category.

- L263: Instead of "overall," I would say "on average" that all your ponds were a $CO_2$ source, as you have also measured a few negative $CO_2$.

Reply: The change has been implemented in the manuscript as recommended (line 297).

- L265: It would be better to call them $CO_2$ "fluxes" instead of "emissions" since you have measured some negative values.

Reply: Thank you for your suggestions. The change has been implemented in the revised manuscript.

- L270: Both are red lines. Please be more precise in the description. What is the mean value of?

Reply: The change has been implemented in the manuscript to better clarify the graph (line 305).

- Table 2: I don't think the abbreviation T-$FCO_2$ is necessary or correct here. I would rather use the term "$CO_2$ fluxes" over the four columns on the right,

including the unit. Where T-FCO$_2$ and the unit are now, I would put the mean ± SD. I also recommend including the number of observations.

Reply: Thank you for your suggestion; the changes have been implemented in Table 2. However, the table has been moved to the Supplementary Material and renamed Table S1 by suggestion of the reviewer 2.

- Figure 3: In the graph, you used n.s., but in the caption, you explained the abbreviation NS. In the supplementary figures, you wrote that an absence indicates no significance. Be consistent.

Reply: The changes have been implemented throughout the manuscript and supplementary material, and we now consistently use "n.s." to indicate non-significant results in the figures.

- L299: Here, you wrote ".01"; in L302, you wrote "0.01" for p. Be consistent throughout the entire manuscript.

Reply: The p-value notation has been corrected for consistency throughout the manuscript.

- Figure 4: What are the R² values of the linear regression lines?

Reply: The R² values of the linear regression lines have been added to Figure 4 in the revised manuscript (lines 334-335).

- L305: The caption of Figure 4 mentions a dashed line, but I cannot see one in the figure.

Reply: This element was removed in the latest version of the figure. We have now updated the caption accordingly to ensure consistency with the figure.

- Table 3: I miss the p values you mentioned in L302f.

Reply: The p-values mentioned in L302f have now been added to Table 1 in the revised manuscript (line 347).

- Figure 5: I'm not sure what to say about these trend lines. I don't trust them because they look like a point cloud with temperature differences and the influence of the edge effect.

Reply: CO$_2$ emissions are not controlled by temperature alone; rather, the model shows it is important for some of the range of temperature, but also with the combined effect of water content. To confirm that the interaction pattern is not an artefact of the model, we compared the fitted relationship with the raw data distribution. A supplementary figure (Fig Sr1) attached shows that the slopes of CO$_2$ fluxes vs. sediment water content differ across temperature levels in the raw data, consistent with the interaction captured by our GLMM.

[Figure]

Fig Sr1. $CO_2$ fluxes trends based on sediment temperature, with dots coloured and sized according to sediment water content. The figure was elaborated using raw data. Lower values are smaller and yellow, while higher values are larger and purple.

- L345: Sometimes there is a space between the number and "Celsius," and sometimes there isn't. Be consistent throughout the manuscript.

Reply: The errors mentioned have been corrected for consistency throughout the manuscript.

- Table 5: In the text, you always used SD, but in the tables, you used sd. Please be consistent and explain every abbreviation in the table caption. Use the same rounding for all emission values shown here. Since seasonality affects $CO_2$ fluxes in your studied ponds and in most ecosystems outside the tropics, it would be helpful to know what season the reference values were measured.

Reply: All the mentioned changes have been implemented in the manuscript.

- L485: The data cannot be reviewed because they are unavailable. I recommend that the authors make the data easily accessible to everyone, not just upon request, for reasons of repeatability and reusability.

Reply: We thank the reviewer for this comment. The data are currently under embargo because they are part of an ongoing doctoral thesis. However, they will be made freely accessible once the embargo period ends, ensuring full repeatability and reusability at that time.

- The information about the R packages used is missing and can be added to the references or the methods chapter.

  Reply: The R packages used have now been included in the Methods chapter, and their corresponding references have also been provided in the bibliography.

- Why did you use two appendices (A and B)?

  Reply: We followed the egusphere journal's guidelines, which recommend the use of separate appendices (A and B) to organize supplementary material more clearly. However, if this format does not fully align with the journal's preferences or if another structure is recommended, we will be glad to adjust it accordingly.

- Table A2: Could you add more lines to better separate the categories in the left column and put each variable name in its own row? This would make the table easier to read.

  Reply: We have reformatted the table, now Table A3 in the revised manuscript, by adding additional lines to better separate the categories in the left column.

- Figure S1: Variable names, including units, are sometimes split into two lines. This makes the table difficult to read. Please reconsider this.

  Reply: We believe the reviewer is referring to Table S1, after modification rename as Table S2. In response, we have reformatted the variable names, including their units, so that they now appear in a single line.

- Figure S2: In the manuscript, you called it "Pond ID"; here, on the x-axis, it is "Pond Code." Be consistent. The y-axis is missing a unit.

  Reply: We thank the reviewer for pointing this out. The labels have been corrected for consistency, and the y-axis unit has now been included in Figure S2.

- Figure S3: Please include the number of observations for each box plot.

  Reply: The number of observations for each box plot has now been included in Figure S3.

- Figure S4: There is a typo in the x-axis label: months.

  Reply: The typo in the x-axis label of Figure S4 has been corrected.

- Figure S5. The sentence about the absence of an asterisk can be removed from the caption.

  Reply: The sentence regarding the absence of an asterisk has been removed from the caption of Figure S5.

Thank you for taking the time to review our manuscript. We have carefully considered all of your comments and suggestions and revised the paper accordingly. We sincerely appreciate your thoughtful and constructive feedback, which has greatly helped us improve the clarity and overall quality of the manuscript. Below, we provide a detailed, point-by-point response to each of your comments, with your remarks shown in black and our replies in blue.

**Referee: 2**

Comments to the Author(s)

The manuscript presents two-season field measurements of $CO_2$ emissions from air-exposed sediments in 30 ponds across Mediterranean and temperate climate zones. Based on the significant relationship between hydroperiod lengths and $CO_2$ emissions in Mediterranean ponds in summer, the authors suggest that longer hydroperiods play a critical role in creating temporary conditions for higher $CO_2$ emissions. Using various statistical approaches, they further identified key drivers of sediment $CO_2$ emissions, including temperature and sediment contents of water and carbonate. The key findings from the well-designed study are novel and invite further study to elucidate the large temporal variability in $CO_2$ emissions from ponds, which have been understudied compared to other freshwater systems. Despite the novelty and significance of the key findings, the manuscript shows weakness in linking and interpreting these findings, as well as a lack of detail in several areas, as described below. I hope my comments will help the authors improve the logical flow and clarity of the manuscript.

Reply: Thank you for your thorough suggestions and comments. We have revised the manuscript based on your feedback; please see our responses to your questions below.

General comments

1. Hypotheses and data interpretation

Although hydroperiods and sediment water contents are suggested as the primary controls on sediment $CO_2$ emissions, descriptions across Introduction, Results, and Discussion appear not consistent, and in some cases contradictory.

First, hypotheses (2) and (3) need to provide more interrelated and mechanistic predictions. Higher sediment contents might be influenced more directly by more recent precipitation events (like 1-month or 1-week antecedent precipitation) than the yearly hydroperiod as considered here. Please provide a more detailed explanation of the relationship between hydroperiods and water contents. Any rationale for using hydroperiod rather than other drought indices would also be helpful.

Second, the findings shown in Fig. 4 indicate the significant relationship between hydroperiods and $CO_2$ emissions only for Mediterranean ponds in summer, and the significance appears controlled by a few sites with very long hydroperiods. However, this hydroperiod effect is emphasized too much across the R & D sections, with some of them having inconsistent connotations: for instance, refer to L 275-276 ("Mediterranean ponds exhibited higher air and sediment temperatures, shorter hydroperiods, typically drying in summer. They also showed lower sediment water content, and reduced

macrophyte coverage, consistent with an earlier drying period."). Please check the consistency of descriptions across R & D (sections 4.1 and 4.2 appear to address two separate stories regarding the hydroperiod effect) to provide a more coherent explanation for the relationship between hydroperiods and sediment water contents.

Reply: Thanks you for the comments. Here we provide a detailed response to all the comments in order:

- First, we selected hydroperiod length as a key explanatory variable because it provides a simple, readily measurable proxy for the cumulative effects of the preceding wet phase. Unlike single-point measurements, hydroperiod length integrates a suite of biotic and abiotic processes that occur during inundation (accumulation and transformation of organic carbon, macrophyte growth and senescence, and changes in nutrient loading), which can influence sediment properties and subsequent $CO_2$ dynamics during the dry phase. Therefore, using hydroperiod length allows us to capture these influences on carbon processing that occur before measurements, without requiring extensive, often unavailable, time-series data, such as precipitation data or water table levels. Short-term data as you suggest (e.g., 1-week to 1-month) are more directly reflected by site-specific sediment measurements, such as sediment water content. Although we found a moderate correlation between hydroperiod length and sediment water content (r =.47; Fig. B1), hydroperiod length captures broader conditions that are not completely explained by sediment water content (Fig. Sr2) and the combination of sediment water content and temperature (Fig. Sr3).

[Figure]

Fig Sr2. Relationship between hydroperiod length (months) and sediment water content (%) in ponds from Mediterranean and Temperate regions, separated by season (Blue= Summer and orange=Autumn). Each point represents the mean per pond and season, and the lines show the linear trend with its confidence interval (shaded area).

[Figure]

Fig. Sr3 Relationship between $CO_2$ fluxes and hydroperiod. To examine the drivers of emissions, dots represent the mean $CO_2$ flux per pond and season. Dot size corresponds to sediment water content, and colour indicates sediment temperature, ranging from blue (low) to red (high). Solid lines represent trends for summer, and dashed lines represent trends for autumn.

We have expanded the explanation in the Introduction (lines 90–96) and added a methodological justification in the Methods section (lines 145-147) to clarify this approach for readers.

- Second, we have revised the manuscript to improve clarity and coherence in the description and interpretation of results related to hydroperiod (lines 420-423). We have included modifications along results (lines 332-338) and discussion (lines 462-463) to improve the new revised manuscript.

**2. Realigning paragraphs**

Although the manuscript was easy to follow on a sentence-by-sentence level, the use of very long or several scattered short paragraphs made it difficult to grasp the overall logical structure. In the Introduction, for example, the page-long initial paragraph is followed by five short paragraphs. A thorough revision of the manuscript is recommended to reorganize the long and short paragraphs in accordance with a coherent logical flow.

Reply: We appreciate the reviewer's observation regarding paragraph structure and logical flow. The Introduction and other sections have been thoroughly revised to improve readability and coherence.

**3. Clarity of tables and figures**

There are numerous missing or inaccurate details that could be improved through careful revision. Please refer to the specific comments below.

Reply: We have carefully revised the tables and figures in accordance with the suggestions and notifications received.

Specific comments

- Title: A slight change would enhance clarity: for example, Drivers of $CO_2$ emissions during the dry phase "in" Mediterranean and Temperate ponds or Drivers of $CO_2$ emissions "from" Mediterranean and Temperate ponds "during the dry phase".

  Reply: We have modified the title following the first suggestion, and it now reads: "Drivers of $CO_2$ emissions during the dry phase **in** Mediterranean and Temperate ponds."

- Line (L) 17: sources of carbon (or $CO_2$)?

  Reply: While "sources of carbon" is more general and correct, we agree that given the focus of our study it is more appropriate to specify "$CO_2$" here. The sentence has been revised accordingly (Line 17).

- L 17 "remain largely overlooked": This statement overlooks the decadal research on this topic.

  Reply: We agree that stating "remain largely overlooked" may be too categorical. The sentence has been revised to specify the ecosystem targeted in our study, now reading: "However, $CO_2$ emissions during the dry phases **of ponds** remain underrepresented in global reports, despite growing evidence that climate change-driven shifts in temperature and precipitation are likely to increase the frequency and duration of these dry periods. " (Lines 17-19).

- L 25: "the" interaction

  Reply: The error has been corrected (Line 26).

- L 35-38: Please provide some estimates of $CO_2$ and $CH_4$ emissions from ponds to describe their role more quantitatively.

  Reply: We have now added representative $CO_2$ emission values from the literature as a reference in lines 40-43. $CH_4$ emission values were not included, as the focus of the present study is on $CO_2$ emissions.

- L 85: Please define "hydroperiod length".

  Reply: In this study, hydroperiod refers to the duration of time each pond retained water prior to the dry phase, quantified as the number of months with water surface during the 12-month period preceding the last autumn sampling (conducted between late September and November). We have now included a definition of hydroperiod length in the manuscript for better clarification in lines 94-95 and better explained in section 2.2.2 Hydrological data (lines 144-146).

"Hydroperiod length (i.e., the duration of water presence prior the dry phase in a pond throughout the year)"

- L 96: Without the above-mentioned definition, it is difficult to understand "shorter hydroperiods leading to lower emissions due to reduced sediment water content".

  Reply: We have now explicitly defined hydroperiod length prior to this section (L96) in lines 94-95. We also rephrased the sentence to clarify this point in lines 107-108.

- L 99: Can you illustrate "conservation status" using an example?

  Reply: We thank the reviewer for the suggestion. An example illustrating "conservation status" has been added to the manuscript (lines 111-113), describing some characteristics of well-conserved ponds.

  "better conservation status (e.g., clear water with turbidity < 5 NTU, extensive native emergent vegetation, and ≥ 50% hydrophytic plant cover, particularly vascular submerged species or charophytes covering > 75% of the pond bottom), will exhibit greater $CO_2$ emissions due to increased vegetation senescence during the dry phase."

- L 117: Did 23 sites also include semi-permanent and permanent ponds? In the latter case, the described bare sediment would be contradictory to the definition of permanent ponds (L 110).

  Reply: Yes, they did. We would like to clarify that the classification of ponds as temporary, semi-permanent, or permanent is based on a three-year record. However, as explained in lines 124-128, during the sampling year (2022), extreme drought conditions caused even some semi-permanent and permanent ponds to dry almost completely. This resulted in large areas of exposed sediment, allowing us to measure $CO_2$ emissions under conditions that can occur in more permanent ponds during extreme dry years. We added a sentence in the manuscript to clarify this aspect (lines 127-128).

- L 130 "water presence": Do you mean rainy days or literal water presence in ponds?

  Reply: We refer to the literal presence of water in the ponds, not rainfall. We clarified this point in the manuscript (lines 144-145).

- L 154 (throughout the manuscript): not Chlorophyll a, but chlorophyll a

  Reply: Thank you for noting this. The error has been corrected throughout the revised manuscript.

- L 167: Please provide key details on the chamber design, including the used material, size, ventilation, etc.

  Reply: We used static, reflective chambers with a surface area of 0.075 m² and a total volume of 8 L (diameter 345 mm, height 160 mm). The chambers were made of polypropylene (PP) plastic and covered with aluminium tape, with a

small fan inside at the top to recirculate the air and prevent stratification, but there was no airflow through the chamber ( no additional tubes or pumps were used during the measurements).
We have added the requested details on the chamber design, including material, size, and ventilation in the manuscript (lines 184-191).

- L 174-177: It would provide useful information for assessing the accuracy of sensor data if you compare sensor and additional GC measurements.

    Reply: We thank the reviewer for the suggestion. $CO_2$ samples measured by GC are available for comparison with the sensors; however, as our study focuses on fluxes, the absolute concentration is less critical than the relative changes over time. The manufacturer-stated precision (± 30 ppm) ensures that relative changes in $CO_2$ concentrations are reliable for flux calculations, we followed the methodology of Bastviken et al. (2015), who demonstrated that mini loggers provide cost-efficient and accurate $CO_2$ flux estimates in terrestrial and aquatic environments (see https://doi.org/10.5194/bg-12-3849-2015 for accuracy and reliability details). We have explicitly referred to this study in the manuscript to clarify and support the validity of the methodological approach followed (lines 215-216).
    Due to the request to assess the accuracy of the sensor data, a comparison between the sensor measurements and the GC analyses has been included in the section entitled Note in the editor's decision.

- L 195: Please provide a relevant reference for this carbonation estimation.

    Reply: We have included the reference in the revised manuscript (Heiri et al., 2001; Martinsen et al., 2019)(line 226).

- L 224: How did you test the normal distribution of your datasets?

    Reply: We tested the normality of our datasets using the Shapiro–Wilk test. However, some data did not meet the assumption of normality. Accordingly, we reanalyzed the data using a non-parametric approach (Mann–Whitney test). Despite this adjustment, our results remained unchanged, showing the same significant differences, now confirmed with the appropriate statistical approach. This information has been added in lines 255-258.

- L 265: Are these negative values from partially water-flooded sediments where phytoplankton take up $CO_2$? Please elaborate on the site characteristics and discuss the meaning of these values (if outside measurement error ranges).

    Reply: We observed negative $CO_2$ fluxes only in a few cases, representing approximately 4% of the total fluxes reported (10 out of 249). Typically, there was only one negative measurement per pond, except for one pond (SP044), which showed two. The mean ± SD of these negative fluxes was -257.6 ± 191.3 mg C $m^{-2}$ $d^{-1}$ (N = 10; min = -611.2, max = -1.4; median = −244.7), with no significant difference between seasons (Summer: N = 5, -275.5 ± 259.1 mg C $m^{-2}$ $d^{-1}$; Autumn: N = 5, -239.7 ± 120.2 mg C $m^{-2}$ $d^{-1}$; t = −0.28, df = 8, p = 0.787).

These sites did not show any consistent relationship with the main drivers of $CO_2$ emissions, such as sediment temperature or water content. The magnitude of these negative fluxes is consistent with values reported in other studies using closed chambers. For instance, Keller et al. (2019) reported -324 mg C m$^{-2}$ d$^{-1}$, while Ma et al. (2013) observed -290 and -436 mg C m$^{-2}$ d$^{-1}$ in under-canopy and inter-plant spaces, respectively. Similarly, Koschorreck et al. (2022) found fluxes ranging from -1,440 to 13,620 mg C m$^{-2}$ d$^{-1}$, with negative values representing 6% of all measurements. Since all measurements in our study were conducted using opaque chambers, and we measured fluxes in bare sediments, it is unlikely that these negative values are due to $CO_2$ uptake by residual phytoplankton, plants, or cryptobiotic crusts. Therefore, these negative fluxes most likely reflect physico-chemical processes in the sediments, probably linked to inorganic reactions (Ma et al., 2013; Marcé et al., 2019). This explanation has been incorporated into the Discussion section of the revised manuscript (lines 473-480).

- L 284-286: Please clarify whether you are talking about the proportion of each component based on unit mass of sediment or DOC.

  Reply: We appreciate the reviewer's observation. The comparison refers to the relative proportion of each PARAFAC component within the total fluorescent DOC signal, rather than to values normalized by sediment mass. We have clarified this in the revised manuscript (lines 313-314).

- L 288-293: These sentences are good examples of unnecessary separation mentioned before.

  Reply: The unnecessary separation between sentences has been removed in the revised version.

- L 300: Given the significance of the hydroperiod effect, it would be helpful to elaborate more as to how "the effect of hydroperiod was season-specific and climate-dependent" as displayed in Fig 4.

  Reply: We have clarified the description of the hydroperiod effect in lines 332-338 to better explain how it was season-specific and climate-dependent. Additionally, the corresponding p-values have been added to Table 1 to provide more detailed information.

- L 301: Was the summer trend also significant for the temperate sites?

  Reply: The summer trend was not significant for the Temperate sites. To clarify this point, we have added the corresponding p-values in the revised manuscript in Table 1 (lines 347). When both climatic regions are considered together, the overall trend is significant; however, when analyzed separately, the significant effect is only observed for the Mediterranean ponds.

- L 320-330: In a sense, this part seems secondary but covers the bulk of section 3.2. More space could be saved for more relevant drivers.

Reply: This section (lines 353-360) has been considerably reduced in the revised manuscript in accordance with your comment, to focus more on the most relevant drivers.

- L 354 "all ponds emitted $CO_2$ during the dry phase": This statement is contradictory to the result descriptions (Fig. 2).

Reply: Only a few measurements (10 out of 249) showed negative $CO_2$ fluxes, but these were minor and isolated occurrences, with overall flux patterns indicating $CO_2$ emission across ponds. To more accurate statement, we modify the manuscript in lines 389-390.

- L 357: "shaped" or "was shaped by"?

Reply: We thank the reviewer for this comment. We have retained the active form "shaped" in the manuscript, as it accurately reflects the causal relationship described (line 392).

- L 445: It would help readers to compare the magnitudes of plant uptake vs. $CO_2$ emissions if you provide some literature values estimating plant C uptake.

Reply: We thank the reviewer for this suggestion. While we agree that it would be valuable to provide literature values for plant C uptake to compare with $CO_2$ emissions, such values strongly depend on the macrophyte species and the characteristics of the specific waterbody. Due to this high variability, it was not possible to provide reliable estimates of species-specific $CO_2$ uptake for our study system. However, to give readers a quantitative perspective, we have now included literature-reported ranges of carbon burial in small ponds depending on vegetation cover (Taylor et al., 2019), as well as a comparison of $CO_2$ fluxes measured in bare and vegetated areas of wetlands under both light and dark conditions to assess the potential of aquatic vegetation to offset $CO_2$ emissions (Sharma et al., under revision), and information on net ecosystem exchange (NEE) from Madaschi et al. (2025). These additions are included in lines 491-498.

L 417: Fig 5 shows the generally highest levels of $CO_2$ emissions across the highest temperature ranges.

Reply: We agree with the reviewer. The corresponding correction has been included in the manuscript in line 457.

- L 460 "ponds with more permanent hydroperiod": This is quite confusing, given your descriptions of your sites. Did you mean simply "longer hydroperiod"?

Reply: We have modified the sentence, now reading "longer hydroperiods" (line 515).

- Fig 1 caption: Countries "are"

Reply: The figure caption has been corrected in the revised manuscript (line 122).

- Fig 2: Please complete the vertical axis title with the second parenthesis.

  Reply: The error has been corrected in the revised version of the manuscript.

- Table 2: If this displays the same data as Fig 2, please think about removing or revising it to avoid double presentation.

  Reply:  Table 2 has been moved to the Supplementary Material (Table S1) for readers interested in specific details on $CO_2$ fluxes.

- Fig 4: Please indicate the significance levels for the depicted regressions. It would be easier to find out the significance if only significant regressions were shown as regression lines.

  Reply: The significance levels of the regressions are now included in Table 1 to indicate which relationships are statistically significant, complementing Figure 4.

- Tables 3, 4, 5: Please explain in the caption the abbreviations including SE, df, CL, AIC, BIC, and CI.

  Reply: Explanations of all abbreviations (SE, df, CL, AIC, BIC, and CI) have been added to the captions of all tables required.

- Figure 5: What is ORQ? Are all the depicted trends statistically significant?

  Reply: ORQ (Ordered Quantile normalization) is a data transformation applied to meet the assumptions of normality, This has been added to the Figure 5 caption. The partial trends shown in Figure 5 were evaluated using 95% confidence intervals of model-predicted values (via the R package visreg). While the confidence intervals for the three sediment temperature levels (9.4 °C, 18.2 °C, and 27.7 °C), cross zero, significance is assessed at the model level. In this GLMM,  the interaction between sediment water content and sediment temperature is statistically significant based on the fixed-effects confidence intervals.

- Table 5: What about showing the employed models in a separate column?

  Reply: We have clarified the model used in the table caption for better readability. Since all estimates come from the same model, adding a separate column in Table 3 was deemed necessary for clarity or aesthetics (line 384).

- Table "6" (page 18): Please also correct the unnecessary values below the decimal point.

  Reply: Thank you for noting these issues. The table number has been corrected, and the unnecessary decimal values have been removed in the revised version of the manuscript.

**Note in the editor's decision**

One thing to note about your response to the first reviewer's comment on $CO_2$ sensor calibration: Can you provide some detail as to how you checked the accuracy and consistency of sensor measurements, for instance using reference gases (or standards)? It is my understanding that, in case sensor measurements drift away from validated standard value, post-correction can be applied to sensors that do now allow for calibration.

Reply: We thank the reviewer for raising this point. To evaluate the accuracy of the Sensirion $CO_2$ sensors used in the study, we provide a comparison between sensor readings with gas chromatograph (GC) measurements for two of the sensors (S1 and S2), which were used for all GHG measurements in Spanish ponds.

For raw absolute concentrations, Sensirion sensors underestimated absolute $CO_2$ concentrations by ~7–11% relative to the gas chromatograph the GC–sensor relationship was strongly linear ($R^2$ = 0.96–0.98) but with slopes slightly below 1 (S1 = 0.93; S2 = 0.89).

[Figure]

Because fluxes depend on the rate of concentration change ($\Delta CO_2/\Delta t$), we also compared GC and sensor data after subtracting the air baseline. This correction improved agreement for both sensors, with slopes of 0.97 (S1) and 0.93 (S2). This confirms that the proportional relationship between GC and sensor measurements is strong and that relative changes in $CO_2$ (the basis for flux calculations) are reliably captured.

[Figure]

The sensors were configured following manufacturer recommendations, including forced recalibration (we used 410 ppm), altitude compensation (60 m), atmospheric pressure compensation (1000 mbar), and a fixed measurement interval, ensuring stable operation across deployments. The Arduino script calibration for one-point $CO_2$ calibration was performed using a custom Arduino script (Code attached in supplement for reviewers' revision).

If needed, we can include the evaluation about the accuracy of the Sensirion $CO_2$ sensors used in the study, plots and $R^2$ in Supplemental Material.

**Relevant changes in the revised manuscript**

We incorporated the information suggested by the reviewers and improved the fluency of the revised manuscript.

We have added a new table in the Appendix (Table A1), line 531.

Figure 1 modified accordance to the review suggestion, line 118.

Figure 2 modified with all corrections and suggestions included in line 301.

Figure 4 added the result of the linear mixed-effects model line 343.

Modification of the statistical approach to a non-parametric method (lines 255–258).

Table 2 has been moved to the supplemental material as Table S1.

Renumbering of table numbers due to modifications.

Conclusion section added (lines 523-529).

Colour schemes were adjusted as indicated (Fig. S1, S6, and Fig. 5).

---

## Author Response (AR2)

Thank you for the latest technical corrections. These have been carefully addressed and incorporated to improve the manuscript. Reviewers' comments are shown in black, and our responses are provided in blue.

**Reviewer and editor comments**

• In L17, I recommend changing "reports" to "studies."

Reply: Change incorporated in the revised manuscript.

• In L41, I don't understand what you mean by "similar size." Are you referring to the ponds you studied?

Reply: Yes, this refers to ponds included in our study. To clarify, we have revised the sentence (lines 40-43) as follows:

" For instance, ponds within the size range of our study($< 0.001$ and $0.001–0.01$ km$^2$) reported by Holgerson and Raymond (2016) emitted on average 254 and 422 mg C m$^{-2}$ d$^{-1}$, respectively, whereas exposed pond sediments reported by Keller et al. (2020) range from -73 to 11765 mg C m$^{-2}$ d$^{-1}$."

• I recommend spelling out the abbreviation of the unit NTU in L111.

Reply: abbreviation spelt out in the revised manuscript.

• L138: For 2022, it is (total) annual precipitation, not the "mean" annual precipitation. Additionally, I suggest putting the term "mean" in brackets in L141.

Reply: The changes recommended have been implemented in the revised manuscript.

• L337: The non-significant p value is "=" and not "<". Additionally, I recommend adding the term "summer" somewhere in this sentence to clarify that this statement only applies to summer.

Reply: The p-value notation has been corrected, and the term "summer" has been added to the sentence to clarify that the statement refers specifically to summer (lines 333-335).

"In summer Mediterranean ponds, with longer hydroperiods were associated with increased $CO_2$ emissions ($p < .001$), whereas in Temperate ponds, the trend was inverse but not statistically significant ($p = .13$) (Table 1)."

• L348: To make this sentence easier to understand, I recommend spelling out that "a variable" is temperature.

Reply: We revised the sentence to improve clarity by explicitly stating that the variable is temperature (lines 345-346):

"We also explored the relationship between $CO_2$ fluxes and temperature (annual and 40-year mean). The variable temperature (annual and 40-year mean) displayed a moderately strong inverse correlation with hydroperiod length (Fig. B1).

• L550: In the response letter, you wrote that the data are embargoed, but you can still include the link to the repository here. Otherwise, it will be difficult for people to find your data later...

Reply: The datasets used in this study are currently under embargo but will be made available at https://dataportal.ponderful.eu/dataset/. In the meantime, they can be provided upon reasonable request. We have included the link to the PONDERFUL project repository in the data availability section to ensure that the data will be easily accessible once the embargo ends.

•L 97-113: Please combine two separate paragraphs into one.

Reply: The two paragraphs have been combined into a single paragraph as requested.

• L 161 "TOC-L analyzer": total organic carbon (TOC) analyzer

Reply: Correction incorporated in the manuscript.

• L 252-253: Please move this single-sentence paragraph to the previous one.

Reply: The sentence has been moved to the previous paragraph as requested.

• L 525: "carbon" fluxes?

Reply: The term "carbon" has been added to the manuscript accordingly.

**Relevant changes in the revised manuscript**

Only minor modifications were made to the affiliation list and corresponding author information. The required copyright statement was added to the map image in accordance with the journal's guidelines, and no other changes were made beyond those explicitly requested by the reviewers and the editor.